# Online Functional Tensor Decomposition via Continual Learning for Streaming Data Completion

**Xi Zhang[1]**    **Yanyi Li[1]**    **Yisi Luo[1]***    **Qi Xie[1]**    **Deyu Meng[1,2]**

[1]Xi'an Jiaotong University, Xi'an, China
[2]Pazhou Laboratory (Huangpu), Canton, China
`xi.zhang@stu.xjtu.edu.cn, yyl0605@foxmail.com`
`yisiluo1221@foxmail.com, xie.qi@mail.xjtu.edu.cn, dymeng@mail.xjtu.edu.cn`

## Abstract

Online tensor decompositions are powerful and proven techniques that address the challenges in processing high-velocity streaming tensor data, such as traffic flow and weather system. The main aim of this work is to propose a novel online functional tensor decomposition (OFTD) framework, which represents a spatial-temporal continuous function using the CP tensor decomposition parameterized by coordinate-based implicit neural representations (INRs). The INRs allow for natural characterization of continually expanded streaming data by simply adding new coordinates into the network. Particularly, our method transforms the classical online tensor decomposition algorithm into a more dynamic continual learning paradigm of updating the INR weights to fit the new data without forgetting the previous tensor knowledge. To this end, we introduce a long-tail memory replay method that adapts to the local continuity property of INR. Extensive experiments for streaming tensor completion using traffic, weather, user-item, and video data verify the effectiveness of the OFTD approach for streaming data analysis. This endeavor serves as a pivotal inspiration for future research to connect classical online tensor tools with continual learning paradigms to better explore knowledge underlying streaming tensor data.

## 1   Introduction

In real world, high-dimensional data often exist in a streaming form and are typically modeled as tensor streams (such as traffic flow and video) Yu et al. [2015], Smith et al. [2018]. Tensor streams can be categorized into two types: single-aspect streams and multi-aspect streams. Single-aspect streams, such as traffic flow data represented by the triplet (location, route, and time), grow only along the temporal dimension and are modeled as 3-mode temporal tensor streams. In contrast, multi-aspect streams, such as recommendation system data represented by the triplet (user, movie, and actor), grow along multiple dimensions simultaneously. With the increasing prevalence of streaming data, there is a growing demand for real-time streaming data analysis (e.g., streaming data completion).

Online tensor decomposition Abed-Meraim et al. [2022a] is one of the foremost methods to address the streaming data completion problem by exploiting the potential compact structures underlying streaming data within an online optimization framework. However, decomposing tensor streams would lead to high computational costs owing to the significant growth in their volume over time. Also, dynamically capturing the internal latent properties of tensor streams, such as spatial-temporal continuity, poses difficult challenges. To address these challenges, various online tensor decomposition methods have been developed to handle the streaming data completion problem. These

---

*Corresponding author.

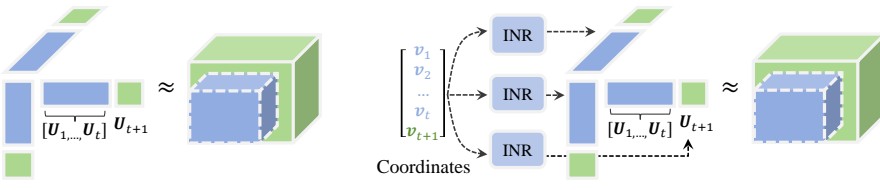

| (a) Classical online CP decomposition | (b) Proposed online functional CP decomposition |

Figure 1: (a) Illustration of the classical online tensor CP decomposition paradigm. (b) The proposed online functional tensor decomposition paradigm for streaming tensor data. Our approach transforms the static online optimization algorithm into a more dynamic continual learning paradigm, which learns knowledge from streaming data by using the CP decomposition parameterized by INRs.

methods can be categorized into CP decomposition-based methods Lee and Shin [2021], Zhong et al. [2021], Ahn et al. [2021], Abed-Meraim et al. [2022b], Liu et al. [2022], Tucker decomposition-based methods Jang and Kang [2023], tensor SVD (t-SVD)-based methods Zhang et al. [2016], Wu [2022], Gilman et al. [2022], Yi et al. [2022], and other tensor network decomposition-based approaches (such as tensor train (TT) decomposition Le et al. [2024]). As recent symbolic works along this line, Ahn et al. [2021] proposed a streaming tensor factorization with an attention-based temporal regularization for streaming data completion. Abed-Meraim et al. [2022b] proposed a scalable online CP algorithm that efficiently estimates low-rank components from streaming tensors. Yi et al. [2022] proposed an online algorithm based on the t-SVD, which can efficiently capture the principal information of tensor by incrementally updating a much smaller sketch. These methods enable real-time analysis of tensor streams and achieve promising performances for streaming data completion. Nonetheless, current online tensor decomposition methods mostly consider optimizing the factor matrices/tensors of the tensor decomposition using discrete online optimization rules, which may limit the capability for dynamic data structure modeling and spatial-temporal correlation excavation.

In this work, we propose a novel online functional tensor decomposition (OFTD) method for streaming data completion (see Fig. 1(b) for quick view). OFTD represents the spatial-temporal streaming data as a continuous function using CP tensor decomposition parameterized by implicit neural representations (INRs) Sitzmann et al. [2020], Mildenhall et al. [2021], which map an arbitrary spatial-temporal coordinate to the corresponding tensor value through deep neural networks. Instead of optimizing the factor matrices of tensor decomposition, OFTD optimizes the learnable weights of factor INRs during online optimization, which could better explore the complex dynamic structure and global spatial-temporal correlations of streaming data through a deep functional tensor representation. OFTD allows for natural characterizations of continually expanded streaming data by simply adding new coordinates into the INR network during online optimization. Although the tensor stream is continually growing, we maintain a constant number of optimization parameters by keeping the network structure unchanged.

Nevertheless, the incorporation of deep neural networks leads to the possible issue of forgetting the learned historical knowledge when fitting new data. Hence, to enable effective continual learning for processing streaming tensors using OFTD, we make a basic attempt by employing the memory replay method, which utilizes a part of historical data during online optimization, to avoid forgetting. We develop a regret bound of OFTD from the perspective of the loss (i.e., forgetting) of historical knowledge, which theoretically shows that the OFTD model tends to forget the knowledge that is distant from the new tensor data due to the local continuity property of INRs. Hence we correspondingly design a memory buffer with a long-tail distribution (i.e., storing more data at more distant positions), which substantially enhances the performance for streaming data completion. Consequently, OFTD delivers superior performances against traditional online tensor decomposition methods, showcasing its strong ability for streaming data analysis. We summarize the main contributions of this work as follows:

- We introduce a novel online functional tensor decomposition (termed OFTD) method for streaming data completion. Our method employs the CP decomposition parameterized by INRs to learn a spatial-temporal continuous function, which enables a concise representation of streaming tensor data by simply incorporating new coordinates into the INR during online optimization. Furthermore, our method effectively captures spatial-temporal continuity and low-dimensional compact structures of streaming data through functional representation.

- Our approach transforms the static online algorithm for streaming data completion into a more dynamic continual learning paradigm, i.e., the INRs are expected to fit new data without forgetting previous tensor knowledge. To achieve this, we theoretically develop a regret bound for OFTD, which guides us in designing a long-tail replay continual learning method tailored for OFTD.

- We apply OFTD to single-aspect (i.e., temporal evolution) and multi-aspect (spatial and temporal evolutions) streaming tensor completion. Extensive experiments on real-world datasets show the superiority of OFTD over state-of-the-art online tensor decomposition methods.

## 2 Related Work

### 2.1 Online Tensor Decomposition

Online analysis algorithms for streaming tensors have been widely developed in recent years Najafi et al. [2019], Qian et al. [2021], Hu et al. [2022]. CP decomposition is one of the mostly considered methods for streaming tensor completion. For instance, Minh-Chinh et al. [2016] proposed a two-stage CP decomposition algorithm to perform streaming data completion for third-order tensors. Lee and Shin [2021] proposed a robust method for tensor streams by integrating CP factorization, outlier removal, and temporal-pattern detection to enable accurate online prediction. Zhong et al. [2021] proposed a window-based dynamic streaming tensor analysis method using CP decomposition. Yang et al. [2023] proposed a distributed streaming tensor completion method for multi-aspect streaming tensor. More CP-based online tensor decomposition methods can be found in Abed-Meraim et al. [2023], Mardani et al. [2015], Kasai [2019], Song et al. [2017], Nimishakavi et al. [2018], Du et al. [2018], Xiao et al. [2018]. These online tensor methods explicitly optimize factor matrices of the tensor decomposition using discrete online optimization rules. Recently, Fang et al. [2021] proposed a deep neural network-based streaming Bayesian tensor factorization method, which could better capture complicated nonlinear interactions in data. Different from this method, our OFTD employs INRs to model streaming data as a spatial-temporal continuous function, which allows for natural characterization of continually expanded data by adding new coordinates into INRs during online optimization, while OFTD could also capture nonlinear interactions of data through the deep representation of INR.

### 2.2 Implicit Neural Representation

In recent years, INR has attracted widespread attention for their ability to continuously and implicitly represent signals through neural networks Sitzmann et al. [2020], Mildenhall et al. [2021]. Compared to traditional discrete representations, INR offers advantages such as memory efficiency, continuous trajectory modeling, and analytical computations of higher-order derivatives. Typical applications of INR include 3D shape and scene reconstruction Atzmon and Lipman [2020], Jiang et al. [2020], view synthesis Mildenhall et al. [2019], solving differential equations Chen et al. [2023], and images/video processing Chen et al. [2021], Skorokhodov et al. [2021]. For instance, the well-known NeRF Mildenhall et al. [2021] represents a scene as a continuous 5D function and optimizes neural radiance fields to synthesize novel views. Sitzmann et al. [2020] introduced the SIREN, which leverages periodic activation functions to achieve high-fidelity representation of natural signals using INR. Chen et al. [2021] proposed the local implicit image function, which encodes images as continuous functions and enables arbitrary-resolution image reconstruction. Currently, INR has become an increasingly popular research direction in mainstream AI fields Zhao et al. [2024], Shi et al. [2024], Li et al. [2024]. To our knowledge, the proposed method should be the first work to incorporate INR into the online tensor decomposition framework.

## 3 Proposed Method

### 3.1 Preliminaries

Basic notations are shown in Table 1. We use $[\![\cdot]\!]$ to denote the Kruskal operator Kolda and Bader [2009]. The $\circ$ and $\otimes$ respectively denote the outer product and the Hadamard product. The $\|\cdot\|_F$ denotes the Frobenius norm.

Table 1: Notations used in this paper.

| Notations | Definitions |
|---|---|
| $x, \mathbf{x}, \mathbf{X}, \mathcal{X}$ | Scalar, vector, matrix, and tensor |
| $\mathcal{X}_t \in \mathbb{R}^{I_1^t \times I_2^t \times \cdots \times I_N^t}$ | An $N^{th}$-order streaming tensor |
| $\mathbf{X}_{(i,:)}, \mathbf{X}_{(:,i)}$ | The $i$-th row or the $i$-th column of $\mathbf{X}$ |
| $\mathcal{X}_{(i_1, i_2, \ldots, i_N)} \in \mathbb{R}$ | The $(i_1, i_2, \ldots, i_N)$-th element of $\mathcal{X}$ |
| $\mathbf{U}^{(n)} \in \mathbb{R}^{I_n \times r}$ | The $n$-th factor of CP decomposition |
| $[N] \in \mathbb{Z}^N$ | The vector $[N] \triangleq (1, 2, \cdots, N)^T$ |

**Definition 1** (CP Decomposition Kolda and Bader [2009]). *Given an $N^{th}$-order tensor $\mathcal{X} \in \mathbb{R}^{I_1 \times I_2 \times \cdots \times I_N}$, its CP decomposition is the representation using $N$ factor matrices $\left\{ \mathbf{U}^{(n)} \right\}_{n=1}^{N}$ sharing the same number of columns as follows:*

$$\mathcal{X} = \left[\!\left[ \left\{ \mathbf{U}^{(n)} \right\}_{n=1}^{N} \right]\!\right] \triangleq \sum_{i=1}^{r} \mathbf{U}^{(1)}_{(:,i)} \circ \cdots \circ \mathbf{U}^{(N)}_{(:,i)}, \tag{1}$$

*where the factor matrix $\mathbf{U}^{(n)} \in \mathbb{R}^{I_n \times r}$. The smallest integer $r$ satisfying (1) is referred to as the CP-rank of $\mathcal{X}$.*

**Definition 2** (Streaming tensor sequence). *A sequence of $N^{th}$-order tensors $\{\mathcal{X}_t\}$ is called streaming tensor sequence if for any $t \in \mathbb{Z}^+, \mathcal{X}_t \subseteq \mathcal{X}_{t+1}$ [2]. The $t$ grows with time, and $\mathcal{X}_t$ is called the snapshot tensor taken at time $t$.*

**Definition 3** (Temporal tube). *Given two successive tensors $\mathcal{X}_{t-1} \in \mathbb{R}^{I_1^{t-1} \times \cdots \times I_N^{t-1}}$ and $\mathcal{X}_t \in \mathbb{R}^{I_1^t \times \cdots \times I_N^t}$ derived from a streaming tensor sequence $\{\mathcal{X}_t\}$, the coming data (i.e., temporal tube) at time $t$ can be represented by $\mathcal{Y}_t = \mathcal{X}_t \setminus \mathcal{X}_{t-1}$, which has the same size as $\mathcal{X}_t$, with entries given by:*

$$(\mathcal{Y}_t)_{(i_1, \ldots, i_N)} = \begin{cases} (\mathcal{X}_t)_{(i_1, \ldots, i_N)} & \text{if } \exists\, I_n^{t-1} < i_n \leq I_n^t, \\ 0 & \text{otherwise.} \end{cases}$$

### 3.2 Problem Formulation

Given a streaming tensor sequence $\{\mathcal{X}_t\}$ with missing entries, we aim to recover the missing data in the current snapshot $\mathcal{X}_t$. Since $\mathcal{X}_{t-1} \subseteq \mathcal{X}_t$, and we have handled $\mathcal{X}_{t-1}$ in previous time steps, the problem is equivalent to completing the elements in $\mathcal{Y}_t = \mathcal{X}_t \setminus \mathcal{X}_{t-1}$. Tensor streams often exhibit low-rank properties, making tensor decomposition models, such as the CP decomposition, suitable for modeling streaming tensors. The optimization problem at time $t$ is typically formulated as

$$\min_{\left\{ \mathbf{U}_t^{(n)} \right\}} \left\| \mathcal{P}_t \otimes \left( \mathcal{Y}_t - \left[\!\left[ \left\{ \mathbf{U}_t^{(n)} \right\}_{n=1}^{N} \right]\!\right] \right) \right\|_F + \mathcal{R}\left( \left\{ \mathbf{U}_t^{(n)} \right\} \right), \tag{2}$$

where $\left\{ \mathbf{U}_t^{(n)} \right\}$ denotes the set of tensor factors, $\mathcal{P}_t$ is a binary tensor representing missing and observed entries of $\mathcal{Y}_t$, and $\mathcal{R}(\cdot)$ is a regularization term for the factor matrices. For instance, most tensor streams exhibit continuity (i.e., smoothness) over the spatial and time dimensions, e.g., the current pollutant measurement is similar to those of the previous and next 10 minutes. Hence, smooth regularizations such as the total variation Pragliola et al. [2023] can be employed. However, in this work we do not impose an explicit regularization $\mathcal{R}(\cdot)$. Instead, the proposed OFTD implicitly captures the spatial-temporal smoothness of data through the implicit smoothness of INRs (see Lemma 1).

### 3.3 Online Functional Tensor Decomposition

In this section, we give the detailed formulation of the proposed online functional tensor decomposition for streaming data completion.

---

[2] We use set notations (e.g., $\subseteq$ and $\setminus$) for tensors by simply viewing a tensor as a set containing its elements.

**Functional Tensor Decomposition**    Before introducing the online algorithm, we first introduce the batch functional tensor decomposition (FTD) (i.e., off-line setting), which uses INRs to parameterize factor matrices of the CP decomposition. Specifically, for the $n$-th CP factor matrix $\mathbf{U}_t^{(n)} \in \mathbb{R}^{I_n^t \times r}$, we use an INR Sitzmann et al. [2020] to parameterize it. Such INR is a multilayer perceptron (MLP) $f_{\Theta_n}(\cdot) : \mathbb{R} \to \mathbb{R}^r$ with parameters $\Theta_n$, which takes a coordinate $v \in \mathbb{R}$ as input and returns a vector:

$$f_{\Theta_n}(v) = \mathbf{W}_d(\sigma(\mathbf{W}_{d-1} \cdots \sigma(\mathbf{W}_1 v))) \in \mathbb{R}^r, \tag{3}$$

where $\Theta_n \triangleq \{\mathbf{W}_i\}_{i=1}^d$ are weight matrices with $\mathbf{W}_i \in \mathbb{R}^{r \times r}$ for $i = 2, 3, \ldots, d$ and $\mathbf{W}_1 \in \mathbb{R}^{1 \times r}$. $\sigma(\cdot) = \sin(\omega_0 \cdot)$ is the sine activation function that is more effective for INR Sitzmann et al. [2020]. To generate the factor matrix $\mathbf{U}_t^{(n)}$, we consider the parallel notation for $I_n^t$ input coordinates:

$$f_{\Theta_n}(\mathbf{v}) \triangleq \left( f_{\Theta_n}(\mathbf{v}_{(1)}), \ldots, f_{\Theta_n}(\mathbf{v}_{(I_n^t)}) \right)^T \in \mathbb{R}^{I_n^t \times r},$$

where $\mathbf{v} = [I_n^t] \triangleq (1, 2, \ldots, I_n^t)^T$ represents the input coordinate vector and $\mathbf{v}_{(i)}$ denotes the $i$-th element of the vector $\mathbf{v}$. We use this output matrix $f_{\Theta_n}(\mathbf{v})$ (i.e., $f_{\Theta_n}([I_n^t])$) as the factor matrix $\mathbf{U}_t^{(n)}$ of the CP decomposition and then obtain the FTD representation of an $N^{th}$-order streaming tensor $\mathcal{X}_t$ as

$$\mathcal{X}_t = \left[\!\left[ \{ f_{\Theta_n}([I_n^t]) \}_{n=1}^N \right]\!\right] \triangleq \sum_{i=1}^r f_{\Theta_1}([I_1^t])_{(:,i)} \circ \cdots \circ f_{\Theta_N}([I_N^t])_{(:,i)}, \tag{4}$$

where $f_{\Theta_n}([I_n^t])_{(:,i)}$ denotes the $i$-th column of the factor matrix $f_{\Theta_n}([I_n^t])$. We have hence used $N$ INRs $\{ f_{\Theta_n}(\cdot) \}_{n=1}^N$ to parameterize the factor matrices of the CP decomposition.

The FTD in (4) enjoys two potential advantages for streaming data analysis. First, the functional representation naturally allows us to model streaming data in an efficient way by simply adding new coordinates into the model (i.e., by expanding the coordinates $[I_n^t]$) during online optimization. With the expanding of coordinates over time, the output of INR (i.e., the factor matrix of CP decomposition) correspondingly increases in size to fit the larger size of the new data stream. Second, the FTD is effective for modeling real-world tensor data because the CP decomposition characterizes the intrinsic low-dimensional structure of the tensor, while the INRs capture the tensor spatial-temporal smoothness (see Lemma 1) to better recover the unobserved entries.

**Online Functional Tensor Decomposition**    We now introduce the proposed OFTD. We mainly describe the multi-aspect setting, and the single-aspect problem can be seen as a special case of the multi-aspect problem. In the online setting, a new tensor $\mathcal{X}_t \in \mathbb{R}^{I_1^t \times I_2^t \cdots \times I_N^t}$ arrives at each time $t$ with missing values. We dynamically update the FTD model to accommodate the incremental growth of the streaming tensor. Our streaming tensor completion algorithm consists of the initialization stage and the online update stage.

*Initialization Stage:* Given an initial streaming tensor $\mathcal{X}_1 \in \mathbb{R}^{I_1^1 \times I_2^1 \times \cdots \times I_N^1}$, we optimize the following objective function based on the FTD representation (4):

$$\min_{\{\Theta_n\}} \left\| \mathcal{P}_1 \otimes \left( \mathcal{Y}_1 - \left[\!\left[ \{ f_{\Theta_n}([I_n^1]) \}_{n=1}^N \right]\!\right] \right) \right\|_F^2, \tag{5}$$

where $\mathcal{P}_1$ is the initial mask, $\mathcal{Y}_1 = \mathcal{X}_1$ is the initial observed tensor, $\Theta_n$ are learnable parameters of the INR $f_{\Theta_n}(\cdot)$, and $[I_n^1]$ denotes the coordinates of the tensor $\left[\!\left[ \{ f_{\Theta_n}([I_n^1]) \}_{n=1}^N \right]\!\right]$ at dimension $n$. To address the optimization problem, we use gradient descent-based methods (e.g., the Adam optimizer) to update the INR parameters $\{\Theta_n\}_{n=1}^N$ with the loss (5).

*Online Stage:* In the online update stage, a new tensor $\mathcal{X}_t \in \mathbb{R}^{I_1^t \times \cdots \times I_N^t}$ ($t \geq 2$) comes at each time point and we need to update the online FTD model to accommodate the growth of streaming tensor sizes. We take the update of the $n$-th dimension as an example. At time $t$, the streaming tensor size grows by the scale of $I_n^t - I_n^{t-1}$ along the $n$-th dimension. We adapt to the new tensor size by simply adding $I_n^t - I_n^{t-1}$ coordinates to the corresponding factor INR $f_{\Theta_n}(\cdot) : \mathbb{R} \to \mathbb{R}^r$. Specifically, suppose that the coordinate vector at the last time point is $[I_n^{t-1}] \triangleq (1, \cdots, I_n^{t-1})^T$, then the new coordinate vector at the time point $t$ is $[I_n^t] \triangleq (1, \cdots, I_n^t)^T$. Correspondingly, the new factor matrix of the CP decomposition along the dimension $n$ at the time point $t$ is obtained by

$$f_{\Theta_n}([I_n^t]) \triangleq \left( f_{\Theta_n}(1), f_{\Theta_n}(2), \ldots, f_{\Theta_n}(I_n^t) \right)^T \in \mathbb{R}^{I_n^t \times r}.$$

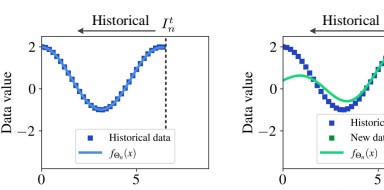 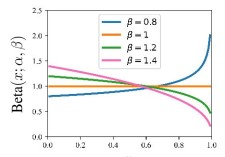 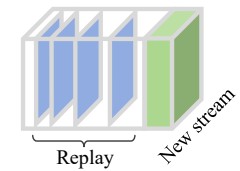

(a) Forgetting behavior of INR        (b) Long-tail memory replay

Figure 2: (a) Illustrations of the forgetting behavior of INR for streaming data completion. (a)-Left: The INR function $f_{\Theta_n}(x)$ (blue line) fits the observed data (blue scatters) well. (a)-Right: When continually fitting the new data (green scatters) without memory buffer, the trained INR $f_{\Theta_n}(x)$ (green line) tends to forget the previously learned knowledge (blue scatters) that is distant from the current time position $I_n^t$ due to the continuity of INR. (b)-Left: Beta distribution with different parameters $\beta$ and fixed $\alpha = 1$. We adopt the long-tail Beta distribution with $\beta > 1$ to construct the memory buffer ((b)-Right) that adapts to the forgetting behavior of INR.

We illustrate the dynamic growth of the coordinates vector $[I_n^t]$ and the corresponding OFTD model in Fig. 1(b). Based on the enlarged OFTD model, we consider the following optimization problem during the online stage $t$:

$$\min_{\{\Theta_n\}} \left\| \mathcal{P}_t \otimes \left( \mathcal{Y}_t - \left[\!\left[ \{f_{\Theta_n}([I_n^t])\}_{n=1}^N \right]\!\right] \right) \right\|_F^2. \tag{6}$$

Similarly, we use the Adam optimizer to update the INR parameters $\{\Theta_n\}_{i=1}^N$ with the loss (6) at each time point $t$. Here, the observed data $\mathcal{Y}_t = (\mathcal{X}_t \setminus \mathcal{X}_{t-1}) \in \mathbb{R}^{I_1^t \times \cdots \times I_N^t}$ contains the new data at time $t$, i.e., the INRs $\{f_{\Theta_n}(\cdot)\}_{n=1}^N$ are optimized to fit the new data in $\mathcal{X}_t \setminus \mathcal{X}_{t-1}$, with initialization weights $\{\Theta_n\}$ being taken from the last time point (i.e., the continual learning). Such online optimization strategy is reasonable since it is impractical to use all historical data at each time point, as this would result in large computational costs. Therefore, we consider only fitting the new data to optimize the INRs in each time point $t$. For single-aspect streams, we continually expand the temporal coordinates of OFTD, while for multi-aspect streams, we need to continually expand the coordinates in all dimensions $n = 1, \cdots, N$. Given a total time $T$, the completion result is obtained by the FTD model $\left[\!\left[ \{f_{\Theta_n}([I_n^T])\}_{n=1}^N \right]\!\right] \in \mathbb{R}^{I_1^T \times \cdots \times I_N^T}$ after $T$ optimization steps under the continual learning manner.

Notably, the streaming data completion using OFTD becomes a classical continual learning paradigm, which learns the INR weights $\{\Theta_n\}$ from a continuous stream of information, with such information becoming progressively available over time Parisi et al. [2019]. Nevertheless, OFTD may encounter forgetting of historical data during online optimization, which we will analyze and address next.

### 3.4  Theoretical Analysis

We interpret two insights of our method, i.e., the spatial-temporal continuity of OFTD brought from INR, and the regret bound of the OFTD model that reveals its forgetting behavior, which motivates us to design memory replay to alleviate forgetting. First, we show that our OFTD method preserves the tensor spatial-temporal smoothness from the Lipschitz smooth perspective.

**Lemma 1** (Lipschitz smooth bound for FTD). *Let the tensor $\mathcal{X}_t \in \mathbb{R}^{I_1^t \times I_2^t \times I_3^t}$ satisfy FTD (4), where each factor function $f_{\Theta_n}(\cdot) : \mathbb{R} \to \mathbb{R}^r$ $(n = 1, 2, 3)$ is an INR formulated as in (3) with activation function $\sigma(\cdot) = \sin(\omega_0 \cdot)$. Assume that each element of the weight matrix $\mathbf{W}$ in (3) follows i.i.d. $\mathcal{N}(0, w^2)$. Then for any $\delta \in (0, 1)$, any spatial coordinates $\mathbf{v}, \mathbf{v}' \in \mathbb{Z}^2$, where $\mathbf{v} = (\mathbf{v}_{(1)}, \mathbf{v}_{(2)}), \mathbf{v}' = (\mathbf{v}'_{(1)}, \mathbf{v}'_{(2)})$ and any temporal coordinates $k, k' \in \mathbb{Z}$, with probability at least $1 - \delta$ the following Lipschitz smoothness holds for $\mathcal{X}_t$:*

$$
\begin{aligned}
|(\mathcal{X}_t)_{(\mathbf{v}_{(1)}, \mathbf{v}_{(2)}, k)} - (\mathcal{X}_t)_{(\mathbf{v}'_{(1)}, \mathbf{v}'_{(2)}, k)}| &\underbrace{\leq C_1 \|\mathbf{v} - \mathbf{v}'\|_{l_1}}_{\text{Spatial smooth}}, \\
|(\mathcal{X}_t)_{(\mathbf{v}_{(1)}, \mathbf{v}_{(2)}, k)} - (\mathcal{X}_t)_{(\mathbf{v}_{(1)}, \mathbf{v}_{(2)}, k')}| &\underbrace{\leq C_1 |k - k'|}_{\text{Temporal smooth}},
\end{aligned}
\tag{7}
$$

*where $C_1 = \omega_0^{3d-3}(2wr^2 + w \ln \frac{3d}{\delta})^{3d} \max(I_1^t I_2^t, I_1^t I_3^t, I_2^t I_3^t)$ is a Lipschitz constant.*

Lemma 1 shows that the FTD model preserves the spatial-temporal continuity of the estimated tensor $\mathcal{X}_t$ (i.e., elements that are closer to each other are more likely to share similar structures). Such continuity benefits the OFTD model by learning a continuous and robust spatial-temporal function that enables more accurate completion results. The smooth bound is related to several factors (such as $\omega_0$ and $w$). We experimentally evaluate such relationships in supplementary.

Based on Lemma 1, we present a regret bound of OFTD regarding the loss of historical information when fitting new data streams, which reveals the forgetting behavior of INR. Such forgetting exhibits a unique characteristic—it is position-dependent due to the spatial-temporal continuity of the OFTD model (see Fig. 2).

**Theorem 1** (Regret bound for online FTD). *Denote the OFTD model learned at the historical time point $t$ by $\mathcal{X}_t = \left[\!\left[ \{ f_{\Theta_n}([I_n^t]; t) \}_{n=1}^3 \right]\!\right] \in \mathbb{R}^{I_1^t \times I_2^t \times I_3^t}$, where $f_{\Theta_n}([I_n^t]; t)$ denotes the $n$-th factor function at time $t$. Assume that*

- *The OFTD model learned at the new time point $t+1$ using (6) is invariant at the boundary [3] $(I_1^t, I_2^t, I_3^t)$, i.e., $f_{\Theta_n}(I_n^t; t+1) = f_{\Theta_n}(I_n^t; t)$ $(n = 1, 2, 3)$.*

- *Each element of the weight matrix of the $d$-layer INRs $\{ f_{\Theta_n}(\cdot) \}$ follows i.i.d. $\mathcal{N}(0, w^2)$ with sine activation function $\sigma(\cdot) = \sin(\omega_0 \cdot)$. The $\ell_1$-norm of derivative of each factor INR $\| f'_{\Theta_n}(x) \|_{\ell_1}$ is bounded by $\kappa > 0$.*

*Then for any $\delta \in (0, 1)$ and any historical position $(i_1, i_2, i_3)$ $(i_n \leq I_n^t)$, the following regret bound between the new OFTD model $\mathcal{X}_{t+1} = \left[\!\left[ \{ f_{\Theta_n}([I_n^{t+1}]; t+1) \}_{n=1}^3 \right]\!\right] \in \mathbb{R}^{I_1^{t+1} \times I_2^{t+1} \times I_3^{t+1}}$ (Take an example $I_n^{t+1} = I_n^t + 1$) and the old OFTD model $\mathcal{X}_t$ holds with probability at least $1 - \delta$:*

$$|(\mathcal{X}_{t+1})_{(i_1, i_2, i_3)} - (\mathcal{X}_t)_{(i_1, i_2, i_3)}| \leq C_2 \max_n (I_n^t + 1 - i_n), \qquad (8)$$

*where $C_2 = 6(\eta^{3d} \omega_0^{3d-3} + \kappa \eta^{2d} \omega_0^{2d-2}) \max(I_1^t I_2^t, I_1^t I_3^t, I_2^t I_3^t)$ and $\eta = 2wr^2 + w \ln \frac{3d}{\delta}$.*

Theorem 1 shows that the regret bound (i.e., the degree of forgetting) is proportional to the positional distance $(I_n^t + 1 - i_n)$ between the considered point $(i_1, i_2, i_3)$ and the new data stream position $I_n^t + 1$. This indicates that information that is more distant from the new data stream is more likely to be forgotten (see Fig. 2(a)). To alleviate the forgetting, we design a long-tail memory replay continual learning method.

### 3.5 Continual Learning via Memory Replay

OFTD transforms the classical online tensor decomposition into a continual learning paradigm, which expects to use the INRs to fit new data without forgetting historical data. To alleviate forgetting, we design a long-tail memory buffer that utilizes a part of historical data when fitting new data. Theorem 1 shows that more distant information is more likely to be forgotten. Thus we consider the long-tail Beta distribution (see Fig. 2(b)) to construct a memory buffer, which stores more data that is distant from the new data stream. For multi-aspect streams, the memory buffer $\mathcal{M}_t \in \mathbb{R}^{I_1^t \times \cdots \times I_N^t}$ at the time $t$ is constructed through a sampling process on the historical data $\mathcal{X}_{t-1}$ by $(\mathcal{M}_t)_{(i_1, \cdots, i_N)} = (\mathcal{X}_{t-1})_{(i_1, \cdots, i_N)}$ if $(i_1, \cdots, i_N) \in \mathcal{I}_t$, otherwise $(\mathcal{M}_t)_{(i_1, \cdots, i_N)} = 0$, where

$$\mathcal{I}_t = \left\{ (i_1, \ldots, i_N) \mid i_n = \lfloor u_n^j I_n^{t-1} \rfloor, u_n^j \sim \text{Beta}(\alpha, \beta), j = 1, \cdots, J, n = 1, \cdots, N \right\}.$$

Here, $\lfloor \cdot \rfloor$ denotes round-down and the p.d.f. of $\text{Beta}(\alpha, \beta)$ is $\text{Beta}(x; \alpha, \beta) = \frac{\Gamma(\alpha+\beta)}{\Gamma(\alpha)\Gamma(\beta)} x^{\alpha-1}(1-x)^{\beta-1}$, $x \in (0, 1)$. The $\mathcal{I}_t$ is the index set of the memory buffer such that the indexes in $\mathcal{I}_t$ follow a long-tail Beta distribution to store more information that is distant from the new data stream. This is achieved by setting appropriate $\alpha$ and $\beta$ such that $\text{Beta}(\alpha, \beta)$ is a long-tail distribution (see Fig. 2(b)). A total number of $J^N$ indexes in $\mathcal{I}_t$ are selected to construct the memory buffer $\mathcal{M}_t$. For single-aspect streams, we perform sampling on the streaming dimension and store all indexes for other dimensions.

---

[3] This assumption is reasonable since data streams in real-world often exhibit local continuity and hence the learned OFTD models at adjacent time points would be invariant at the boundary $I_n^t$.

Table 2: The NRE results of single-aspect streaming data completion. Average running time during each online update is reported.

| Dataset | Condition | | | Beijing | | | Madrid | | | sensor | | | radar | | | chicago | | | NRE | Time (s) |
|---|---|---|---|---|---|---|---|---|---|---|---|---|---|---|---|---|---|---|---|---|
| SR | 0.1 | 0.2 | 0.3 | 0.1 | 0.2 | 0.3 | 0.1 | 0.2 | 0.3 | 0.1 | 0.2 | 0.3 | 0.1 | 0.2 | 0.3 | 0.1 | 0.2 | 0.3 | | |
| Grouse | 1.532 | 1.504 | 1.254 | 2.978 | 2.884 | 1.647 | 1.276 | 1.113 | 1.073 | 2.263 | 1.086 | 0.899 | 4.125 | 2.033 | 1.861 | 1.005 | 1.003 | 1.002 | 1.697 | 0.0135 |
| Grasta | 1.580 | 1.044 | 1.021 | 1.179 | 1.119 | 1.066 | 1.161 | 1.065 | 0.899 | 1.091 | 1.050 | 1.002 | 1.497 | 1.391 | 1.193 | 1.026 | 1.013 | 0.975 | 1.132 | 0.0357 |
| Petrels | 0.940 | 0.609 | 0.567 | 1.077 | 0.607 | 0.343 | 0.696 | 0.287 | 0.180 | 1.504 | 1.362 | 1.034 | 1.068 | 1.016 | 0.831 | 0.791 | 0.728 | 0.696 | 0.797 | 0.0234 |
| TeCPSGD | 1.025 | 0.985 | 0.738 | 1.317 | 0.912 | 0.842 | 0.891 | 0.201 | 0.145 | 4.583 | 2.683 | 1.807 | 1.670 | 1.254 | 0.933 | 0.838 | 0.753 | 0.730 | 1.239 | 0.0358 |
| OLSTEC | 0.506 | 0.272 | 0.204 | 0.226 | 0.185 | 0.167 | 0.456 | 0.222 | 0.155 | 0.723 | 0.535 | 0.435 | 0.876 | **0.613** | **0.558** | 0.847 | 0.805 | 0.785 | 0.476 | 0.4694 |
| SOFIA | 0.125 | **0.065** | **0.070** | 0.217 | 0.218 | 0.171 | 0.322 | 0.250 | 0.243 | 1.794 | 1.532 | 1.411 | 1.660 | 1.469 | 1.326 | 0.860 | 0.837 | 0.823 | 0.744 | 0.0002 |
| STF | 0.347 | 0.282 | 0.184 | 0.325 | 0.241 | 0.184 | 0.995 | 0.972 | 0.965 | 3.653 | 1.097 | 0.875 | 3.400 | 3.046 | 2.660 | 3.532 | 2.357 | 0.177 | 1.405 | 0.0030 |
| ACP | 0.870 | 0.563 | 0.497 | 0.966 | 0.871 | 0.518 | 1.033 | 1.001 | 0.976 | 1.090 | 1.024 | 1.028 | 1.050 | 1.012 | 0.975 | 1.016 | 1.012 | 1.019 | 0.918 | 0.0022 |
| ATD | 1.365 | 1.303 | 1.221 | 1.548 | 0.784 | 0.252 | 0.368 | 0.195 | 0.156 | 1.701 | 1.034 | 0.636 | 1.633 | 1.628 | 0.891 | 0.928 | 0.782 | 0.736 | 0.953 | 0.0043 |
| OFTD(Ours) | **0.116** | 0.094 | 0.093 | **0.156** | **0.141** | **0.131** | **0.138** | **0.125** | **0.118** | **0.528** | **0.433** | **0.379** | **0.869** | 0.827 | 0.821 | **0.607** | **0.507** | **0.444** | **0.363** | 0.1762 |

Involving the memory buffer during online optimization allows our model to retain historical data in $\mathcal{M}_t$ over time, preventing forgetting. By reformulating the online model (6), the new online optimization model at time $t$ with the memory buffer $\mathcal{M}_t$ is formulated as

$$\min_{\{\Theta_n\}} \left\| \mathcal{P}_t \otimes \left( \mathcal{Y}_t - \left[\!\left[ \{f_{\Theta_n}([I_n^t])\}_{n=1}^N \right]\!\right] \right) \right\|_F^2 + \sum_{\mathbf{v} \in \mathcal{I}_t} \left( \mathcal{P}_{t(\mathbf{v})} \otimes \left( \mathcal{M}_{t(\mathbf{v})} - \left[\!\left[ \{f_{\Theta_n}(\mathbf{v}_{(n)})\}_{n=1}^N \right]\!\right] \right) \right)^2 .$$
(9)

Here, $\mathcal{Y}_t$ contains new data at time $t$ and $\mathcal{M}_t$ includes a part of historical data to avoid forgetting. The more historical data in $\mathcal{M}_t$ (i.e., the larger index set $\mathcal{I}_t$), the more computational costs are needed for optimization. If we use all historical data to construct the memory buffer $\mathcal{M}_t$, then OFTD degrades to the batch FTD method that processes the whole tensor $\mathcal{X}_t$ at each time. We summarize the OFTD algorithm with memory replay in Algorithm A.1 of the Appendix.

Given a tensor of size $I_1 \times \cdots \times I_N$, OFTD consumes $O(mrd \sum_{n=1}^N I_n + r \prod_{n=1}^N I_n)$ in each iteration, where $(mrd \sum_{n=1}^N I_n)$ is the INR complexity, $(r \prod_{n=1}^N I_n)$ is the CP product complexity, $r$ is the CP rank, and $m, d$ are network's width and depth. We further propose a sharing strategy to improve the computational efficiency of OFTD, as detailed in Appendix A.2.

To further improve the effectiveness of OFTD for recovering highly irregular data streams (such as dynamic background and abrupt changes in data streams), we further proposed a temporal online affine regularizer to address this special situation, which is introduced in the Appendix A.3.

## 4 Experiments

We perform numerical experiments for both single-aspect and multi-aspect streaming data completion. We use the normalized reconstruction error (NRE) Ahn et al. [2021] for evaluation. The datasets are summarized in Appendix C.1. We consider the sampling rates (SRs) $0.1, 0.2, 0.3$ (the proportion of observed entries w.r.t. all entries) to perform random missing. Various baselines are included: Grouse Balzano et al. [2010], Grasta He et al. [2012], Petrels Chi et al. [2013], TeCPSGD Mardani et al. [2015], OLSTEC Kasai [2019], SOFIA Lee and Shin [2021], ACP Abed-Meraim et al. [2023], ATD Abed-Meraim et al. [2023], STF Ahn et al. [2021], SIITA Nimishakavi et al. [2018], OnlineSGD (OSGD) Mardani et al. [2015], and GOCPT Yang et al. [2022]. More detailed experimental settings are put in Appendix C.2.

### 4.1 Experimental Results

The quantitative results for single-aspect and multi-aspect streaming data completion are shown in Tables 2 and 3. OFTD attains better NRE results in most cases, showcasing its strong representation abilities and effectiveness for modeling tensor streams. This can be attributed to the compact, low-rank representation of CP decomposition and the expressive power of INRs to model dynamic structures of data streams. The OFTD achieves real-time updating with each online step costing less than 0.2/1.0 (single/multi-aspect) seconds. Also, from Fig. 3 it can be observed that OFTD better reconstructs the temporal curves of tensor streams, showcasing its capability to model complex data structures and preserve the temporal smoothness of the tensor. Overall, OFTD serves as a new state-of-the-art

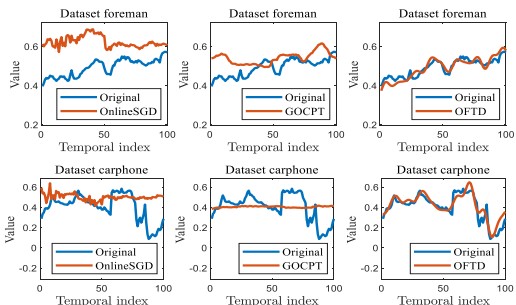

Figure 3: Temporal curves of spatial pixel (SR 0.3).

Table 3: The NRE results of multi-aspect streaming data completion. Average running time during each online update is reported.

| Dataset | foreman | | | carphone | | | YELP | | | Avg. NRE | Time (s) |
|---|---|---|---|---|---|---|---|---|---|---|---|
| SR | 0.1 | 0.2 | 0.3 | 0.1 | 0.2 | 0.3 | 0.1 | 0.2 | 0.3 | | |
| SIITA | 0.257 | 0.252 | 0.249 | 0.557 | 0.556 | 0.464 | 0.415 | 0.368 | 0.340 | 0.384 | 0.020 |
| OSGD | 0.877 | 0.255 | 0.193 | 0.723 | 0.259 | 0.251 | 2.146 | 2.002 | 1.853 | 0.951 | 0.692 |
| GOCPT | 0.146 | 0.140 | 0.140 | 0.168 | 0.153 | 0.153 | 2.055 | 1.312 | 1.291 | 0.618 | 0.462 |
| OFTD | **0.094** | **0.087** | **0.084** | **0.128** | **0.119** | **0.111** | **0.371** | **0.291** | **0.267** | **0.172** | 0.898 |

online method for both single-aspect and multi-aspect streaming data completion. We show more experimental results in Appendix D.

## 4.2 Ablation Study

The ablation studies include the tests for the memory buffer size $J^N$ in Table 4 and the Beta distribution parameter $\beta$ (with $\alpha$ fixed to 1) in Table 5. The memory buffer is effective to alleviate forgetting (compared to $0\%$ in Table 4), thus enhancing performances. However, a large buffer size leads to increased computational costs, and we have set the buffer size to $33\%$ of the whole tensor size in experiments. When the Beta distribution parameter $\beta > 1$, we obtain the desired long-tail memory buffer, resulting in good performance (Table 5) and justifying our memory buffer design. More ablation results include the CP-rank $r$, the usage of INR and its parameters are shown in Appendix D.

Table 4: Ablation study for the memory buffer size (the proportion of buffer size $J^N$ w.r.t. the whole tensor size). A larger buffer leads to improved accuracy while more floating-point operations (FLOPs).

| Dataset | Beijing | | | | Madrid | | | |
|---|---|---|---|---|---|---|---|---|
| Buffer size | 100% | 50% | 33% | 0% | 100% | 50% | 33% | 0% |
| NRE | 0.1280 | 0.1299 | 0.1308 | 0.2903 | 0.1112 | 0.1160 | 0.1182 | 0.3979 |
| FLOPs | 87.97M | 43.99M | 29.33M | 6.03M | 139.42M | 69.73M | 46.5M | 9.09M |

Table 5: Ablation study for the parameter $\beta$ of the Beta distributed memory buffer. When $\beta = 1$ it degrades to the uniform distribution and $\beta > 1$ (or $< 1$) means long-tail (or heavy-tail) memory buffer.

| Dataset | Beijing Madrid (average) | | | | | | | | |
|---|---|---|---|---|---|---|---|---|---|
| $\beta$ | 0.2 | 0.4 | 0.6 | 0.8 | 1 | 1.2 | 1.4 | 1.6 | 1.8 |
| NRE | 0.1273 | 0.1271 | 0.1280 | 0.1264 | 0.1261 | 0.1245 | 0.1239 | 0.1233 | 0.1233 |

## 5 Conclusion

We have proposed a novel streaming data completion method OFTD, which utilizes CP decomposition parameterized by INRs to model tensor streams. Future work can be considered to design other advancing continual learning methods to enable life-long learning of INRs for the streaming data completion problem. For example, we can consider using regularization-based Sun et al. [2023] or gradient projection-based Lin et al. [2022] methods to further boost the performance of OFTD under the continual learning framework. Also, applying the low-rank functional parameterization using INRs paves a novel paradigm for low-rank adaptation (LoRA) of large models, which can be considered in future work.

## Acknowledgments

This work was supported by the National Natural Science Foundation of China (Nos. 124B2029, 62476214), the Major Key Project of PCL (No. PCL2024A06), the Tianyuan Fund for Mathematics of the National Natural Science Foundation of China (No. 12426105), and the National Key Research and Development Program of China (No. 2024YFA1012000).

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

# A The Online FTD algorithm

## A.1 Algorithm Description

---

**Algorithm 1** Online FTD for Streaming Data Completion

---

**Input:** $N^{th}$-order tensor stream $\mathcal{X}_1, \ldots, \mathcal{X}_T$ and masks $\mathcal{P}_1, \cdots, \mathcal{P}_T$; Maximum iteration number $K$;
**Initialization:** Randomly initialize learnable parameters of INRs $\{\Theta_n\}_{n=1}^N$; $k = 0$;

   ▷ (*Initialization Stage*)
1: Construct coordinate vectors $\{[I_n^1]\}_{n=1}^N$;
2: **for** $k \leq K$ **do**
3:    Generate the estimated tensor via the FTD representation $\left[\!\left[\left\{f_{\Theta_n}([I_n^1])\right\}_{n=1}^N\right]\!\right]$;
4:    Compute the loss in (5) using the observed tensor $\mathcal{X}_1$, mask $\mathcal{P}_1$, and the generated tensor using FTD;
5:    Update INR parameters $\{\Theta_n\}_{n=1}^N$ with the loss (5) using the Adam algorithm; Set $k \leftarrow k + 1$;
6: **end for**
   ▷ (*Online Stage*)
7: **while** $t \leq T$ **do**
8:    Construct coordinate vectors $\{[I_n^t]\}_{n=1}^N$ by expanding the previous coordinate vectors; Set $k = 0$;
9:    Construct memory buffer $\mathcal{M}_t$ via Beta distribution;
10:    **for** $k \leq K$ **do**
11:      Generate the estimated tensor via the FTD representation $\left[\!\left[\left\{f_{\Theta_n}([I_n^t])\right\}_{n=1}^N\right]\!\right]$;
12:      Compute the loss in (9) using the observed tensor $\mathcal{X}_t$, mask $\mathcal{P}_t$, memory buffer $\mathcal{M}_t$, and the generated tensor using FTD;
13:      Update INR parameters $\{\Theta_n\}_{n=1}^N$ with the loss (9) using the Adam algorithm; Set $k \leftarrow k + 1$;
14:    **end for**
15:    Set $t \leftarrow t + 1$;
16: **end while**
**Output:** The recovered tensor $\left[\!\left[\left\{f_{\Theta_n}([I_n^T])\right\}_{n=1}^N\right]\!\right]$;

---

## A.2 Computational Complexity and A Sharing Strategy

Here, we propose a sharing strategy to further reduce INR complexity. We share the first $d - 1$ layer weights of the $N$ factor INRs and keep the last layer unshared. This reduces the INR complexity from $O(mrd\sum_{n=1}^N I_n)$ to $O(mr(d-1)\hat{I} + mr\sum_{n=1}^N I_n)$, where $\hat{I} = \max_n I_n$, leading to reduction of running time around 50% without obvious accuracy degradation (Table 6).

Table 6: OFTD (share) reduces running time around 50%.

| Dataset | foreman | | | carphone | | | YELP | | | Time (s) |
|---|---|---|---|---|---|---|---|---|---|---|
| SR | 0.1 | 0.2 | 0.3 | 0.1 | 0.2 | 0.3 | 0.1 | 0.2 | 0.3 | |
| GOCPT | 0.146 | 0.140 | 0.140 | 0.168 | 0.153 | 0.153 | 2.055 | 1.312 | 1.291 | 0.462 |
| OFTD (share) | 0.097 | 0.092 | 0.088 | 0.133 | 0.124 | 0.121 | 0.374 | 0.336 | 0.281 | **0.454** |
| OFTD | **0.094** | **0.087** | **0.084** | **0.128** | **0.119** | **0.111** | **0.371** | **0.291** | **0.267** | 0.898 |

## A.3 Temporal Online Affine Regularizer

To further enhance OFTD for highly irregular data streams, we propose a temporal online affine regularizer to enhance the robustness of OFTD to abrupt changes in temporal streams. We assume that the complete tensor stream $\mathcal{X}_t \in \mathbb{R}^{I_1^t \times I_2^t \times I_3^t}$ is implicitly low-rank. In other words, there exists an explicitly low-rank tensor $[\mathcal{X}_t]_L \in \mathbb{R}^{I_1^t \times I_2^t \times I_3^t}$ and a set of per-frame affine transformations such that the full tensor can be expressed as a transformed version of $[\mathcal{X}_t]_L$. This formulation allows the model to capture geometric or temporal abrupt changes in data streams while preserving the low-rank structure.

Our affine method is inspired by the temporal affine transform introduced in Miao et al. [2024], but adapting this affine transform to online optimization is quite challenging. To address this challenge, we propose to decompose the transformation into structured components, including per-frame translation $(x_t, y_t)$, rotation $\theta_t$, and scaling $s_t$. The translation parameters $\{x_t, y_t\}_{t=1}^{I_3^T}$ are predicted via lightweight neural networks $f_{\Theta_x}(\cdot), f_{\Theta_y}(\cdot) : \mathbb{R}^r \to \mathbb{R}$ with parameters $\Theta_x, \Theta_y$, which take each row of the time-dependent CP factor matrix $f_{\Theta_3}([I_3^t]) \in \mathbb{R}^{I_3^t \times r}$ as input and returns a scalar. Rotation and scaling factors $\{s_t, \theta_t\}_{t=1}^{I_3^T}$ are directly learned as independent parameters for each frame.

For each frame $t$, we construct the affine transformation matrix $\gamma_t \in \mathbb{R}^{2 \times 3}$ as:

$$\gamma_t = \begin{bmatrix} s_t \cos\theta_t & -s_t \sin\theta_t & \cos\theta_t \cdot x_t - \sin\theta_t \cdot y_t \\ s_t \sin\theta_t & s_t \cos\theta_t & \sin\theta_t \cdot x_t + \cos\theta_t \cdot y_t \end{bmatrix}, \tag{1}$$

where $x_t = f_{\Theta_x}(f_{\Theta_3}([I_3^t])_{(t,:)})$ and $y_t = f_{\Theta_y}(f_{\Theta_3}([I_3^t])_{(t,:)})$ are predicted translations, and $s_t$, $\theta_t$ are directly learnable scalars for each frame.

Each pixel location $(i, j)$ is transformed as: $\begin{bmatrix} \hat{i} & \hat{j} \end{bmatrix}^T = \gamma_k \begin{bmatrix} i & j & 1 \end{bmatrix}^T$, and the warped background is obtained via bilinear sampling:

$$\mathcal{X}_{(i,j,k)} = I\left([\mathcal{X}]_L^{(k)},\ (\hat{i}, \hat{j})\right), \tag{2}$$

where $I(\cdot)$ denotes the bilinear interpolation function and $I\left([\mathcal{X}]_L^{(k)},\ (\hat{i}, \hat{j})\right)$ returns the interpolation result of the matrix $[\mathcal{X}]_L^{(k)}$ at the coordinate $(\hat{i}, \hat{j})$.

To validate the effectiveness of the proposed temporal online affine regularizer, we performed streaming data completion on videos "foreman" and "carphone" with sampling rates ranging from 0.1 to 0.3, keeping all other settings the same as those in the main text. The OFTD with temporal online affine regularizer attains better NRE results (see table 7 and fig. 4). This can be attributed to the design that enables frame-wise affine modeling to adapt to temporal changes. The regularizer provides a strong inductive bias by constraining the transformation space to physically consistent temporal trajectories.

Table 7: Ablation study for the affine regularizer in the OFTD model.

| Dataset | foreman | | | carphone | | |
|---|---|---|---|---|---|---|
| SR | 0.1 | 0.2 | 0.3 | 0.1 | 0.2 | 0.3 |
| OFTD w/o affine | 0.134 | 0.121 | 0.114 | 0.146 | 0.123 | 0.119 |
| OFTD w/ affine | **0.114** | **0.104** | **0.099** | **0.134** | **0.119** | **0.112** |

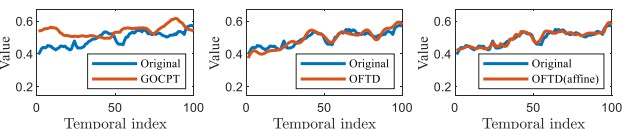

Figure 4: Affine regularizer enhances OFTD for irregular data.

# B  Deferred Proofs

## B.1  Intermediate Lemmas

**Lemma 1** (Lipschitz smooth bound for FTD). *Let the tensor $\mathcal{X}_t \in \mathbb{R}^{I_1^t \times I_2^t \times I_3^t}$ satisfy FTD (4), where each factor function $f_{\Theta_n}(\cdot) : \mathbb{R} \to \mathbb{R}^r$ ($n = 1, 2, 3$) is an INR formulated as in (3) with activation function $\sigma(\cdot) = \sin(\omega_0 \cdot)$. Assume that each element of the weight matrix $\mathbf{W}$ in (3) follows i.i.d. $\mathcal{N}(0, w^2)$. Then for any $\delta \in (0, 1)$, any spatial coordinates $\mathbf{v}, \mathbf{v}' \in \mathbb{Z}^2$, where $\mathbf{v} = (\mathbf{v}_{(1)}, \mathbf{v}_{(2)}), \mathbf{v}' = (\mathbf{v}'_{(1)}, \mathbf{v}'_{(2)})$ and any temporal coordinates $k, k' \in \mathbb{Z}$, with probability at least*

$1 - \delta$ *the following Lipschitz smoothness holds for* $\mathcal{X}_t$:

$$\underbrace{|(\mathcal{X}_t)_{(\mathbf{v}_{(1)},\mathbf{v}_{(2)},k)} - (\mathcal{X}_t)_{(\mathbf{v}'_{(1)},\mathbf{v}'_{(2)},k)}| \leq C_1\|\mathbf{v}-\mathbf{v}'\|_{l_1}}_{\text{Spatial smooth}},$$

$$\underbrace{|(\mathcal{X}_t)_{(\mathbf{v}_{(1)},\mathbf{v}_{(2)},k)} - (\mathcal{X}_t)_{(\mathbf{v}_{(1)},\mathbf{v}_{(2)},k')}| \leq C_1|k-k'|}_{\text{Temporal smooth}}, \tag{3}$$

*where* $C_1 = \omega_0^{3d-3}(2wr^2 + w\ln\frac{3d}{\delta})^{3d}\max(I_1^t I_2^t, I_1^t I_3^t, I_2^t I_3^t)$ *is a Lipschitz constant.*

*Proof.* Since each element of the weight matrix $\mathbf{W} \in \mathbb{R}^{r\times r}$ follows i.i.d. $\mathcal{N}(0, w^2)$, we have the tail bound

$$\mathbb{P}(\|\mathbf{W}\|_{\ell_1} \leq \eta) \geq 1 - \exp(-\eta/w + 2r^2), \tag{4}$$

and hence the probability of all the $3d$ (where $d$ denotes the number of layers for each factor INR) weight matrices of the OFTD model have $\ell_1$-norm less that $\eta$ is at least $1 - 3d\exp(-\eta/w + 2r^2)$. Let $1 - 3d\exp(-\eta/w + 2r^2) = 1 - \delta$ we have $\eta = 2wr^2 + w\ln\frac{3d}{\delta}$. Note that the sine activation function satisfies $|\sin(\omega_0 x)| \leq \omega_0|x|$ ($\omega_0 > 0$). Hence we have $\|f_{\Theta_n}(x)\|_{\ell_1} \leq \eta^d\omega_0^{d-1}|x|$. For any $(i,j,k)$ we have

$$(\mathcal{X}_t)_{(i,j,k)} = \sum_{m=1}^{r} f_{\Theta_1}([I_1^t])_{(i,m)} f_{\Theta_2}([I_2^t])_{(j,m)} f_{\Theta_3}([I_3^t])_{(k,m)}. \tag{5}$$

Then for any $(i,j,k)$ we have

$$\begin{aligned}
\left|(\mathcal{X}_t)_{(i,j,k)} - (\mathcal{X}_t)_{(i',j,k)}\right| &= \left|\sum_{m=1}^{r} \left(f_{\Theta_1}([I_1^t])_{(i,m)} - f_{\Theta_1}([I_1^t])_{(i',m)}\right) f_{\Theta_2}([I_2^t])_{(j,m)} f_{\Theta_3}([I_3^t])_{(k,m)}\right| \\
&\leq \left\|f_{\Theta_1}([I_1^t])_{(i,:)} - f_{\Theta_1}([I_1^t])_{(i',:)}\right\|_{\ell_1} \left\|f_{\Theta_2}([I_2^t])_{(j,:)}\right\|_{\ell_1} \left\|f_{\Theta_3}([I_3^t])_{(k,:)}\right\|_{\ell_1} \\
&= \left\|f_{\Theta_1}(i) - f_{\Theta_1}(i')\right\|_{\ell_1} \left\|f_{\Theta_2}(j)\right\|_{\ell_1} \left\|f_{\Theta_3}(k)\right\|_{\ell_1}.
\end{aligned} \tag{6}$$

The bound on the first term holds

$$\begin{aligned}
\|f_{\Theta_1}(i) - f_{\Theta_1}(i')\|_{\ell_1} &= \|\mathbf{W}_d(\sigma(\mathbf{W}_{d-1}\cdots\sigma(\mathbf{W}_1 i))) - \mathbf{W}_d(\sigma(\mathbf{W}_{d-1}\cdots\sigma(\mathbf{W}_1 i')))\|_{\ell_1} \\
&\leq \eta\|\sigma(\mathbf{W}_{d-1}\cdots\sigma(\mathbf{W}_1 i)) - \sigma(\mathbf{W}_{d-1}\cdots\sigma(\mathbf{W}_1 i'))\|_{\ell_1} \\
&\leq \eta\omega_0\|\mathbf{W}_{d-1}\cdots\sigma(\mathbf{W}_1 i) - \mathbf{W}_{d-1}\cdots\sigma(\mathbf{W}_1 i')\|_{\ell_1} \\
&\cdots \\
&\leq \eta^d\omega_0^{d-1}|i-i'|.
\end{aligned} \tag{7}$$

Thus we have

$$\begin{aligned}
\left|(\mathcal{X}_t)_{(i,j,k)} - (\mathcal{X}_t)_{(i',j,k)}\right| &\leq \|f_{\Theta_1}(i) - f_{\Theta_1}(i')\|_{\ell_1} \|f_{\Theta_2}(j)\|_{\ell_1} \|f_{\Theta_3}(k)\|_{\ell_1} \\
&\leq \eta^d\omega_0^{d-1}|i-i'|\,\eta^d\omega_0^{d-1}|j|\,\eta^d\omega_0^{d-1}|k| \\
&\leq \eta^{3d}\omega_0^{3d-3}I_2^t I_3^t|i-i'|.
\end{aligned} \tag{8}$$

Similarly, we can derive:

$$\begin{cases}
|(\mathcal{X}_t)_{(i,j,k)} - (\mathcal{X}_t)_{(i,j',k)}| \leq \eta^{3d}\omega_0^{3d-3}I_1^t I_3^t|j-j'| \\
|(\mathcal{X}_t)_{(i,j,k)} - (\mathcal{X}_t)_{(i,j,k')}| \leq \eta^{3d}\omega_0^{3d-3}I_1^t I_2^t|k-k'|.
\end{cases} \tag{9}$$

Hence from the triangle inequality one can easily derive that for any spatial coordinates $\mathbf{v}, \mathbf{v}'$ and temporal coordinates $k, k'$ the following bound holds with probability at least $1 - \delta$:

$$\begin{aligned}
|(\mathcal{X}_t)_{(\mathbf{v}_{(1)},\mathbf{v}_{(2)},k)} - (\mathcal{X}_t)_{(\mathbf{v}'_{(1)},\mathbf{v}'_{(2)},k)}| &\leq C_1\|\mathbf{v}-\mathbf{v}'\|_{l_1}, \\
|(\mathcal{X}_t)_{(\mathbf{v}_{(1)},\mathbf{v}_{(2)},k)} - (\mathcal{X}_t)_{(\mathbf{v}_{(1)},\mathbf{v}_{(2)},k')}| &\leq C_1|k-k'|,
\end{aligned} \tag{10}$$

where $C_1 = \omega_0^{3d-3}(2wr^2 + w\ln\frac{3d}{\delta})^{3d}\max(I_1^t I_2^t, I_1^t I_3^t, I_2^t I_3^t)$. The proof is completed. $\square$

## B.2 Proof of Online FTD

**Theorem 1** (Regret bound for online FTD). *Denote the OFTD model learned at the historical time point $t$ by $\mathcal{X}_t = \left[\!\left[ \{f_{\Theta_n}([I_n^t]; t)\}_{n=1}^3 \right]\!\right] \in \mathbb{R}^{I_1^t \times I_2^t \times I_3^t}$, where $f_{\Theta_n}([I_n^t]; t)$ denotes the $n$-th factor function at time $t$. Assume that*

- *The OFTD model learned at the new time point $t+1$ using (7) is invariant at the boundary[4] $(I_1^t, I_2^t, I_3^t)$, i.e., $f_{\Theta_n}(I_n^t; t+1) = f_{\Theta_n}(I_n^t; t)$ ($n = 1, 2, 3$).*
- *Each element of the weight matrix of the $d$-layer INRs $\{f_{\Theta_n}(\cdot)\}$ follows i.i.d. $\mathcal{N}(0, w^2)$ with sine activation function $\sigma(\cdot) = \sin(\omega_0 \cdot)$. The $\ell_1$-norm of derivative of each factor INR $\|f'_{\Theta_n}(x)\|_{\ell_1}$ is bounded by $\kappa > 0$.*

*Then for any $\delta \in (0, 1)$ and any historical position $(i_1, i_2, i_3)$ ($i_n \leq I_n^t$), the following regret bound between the new OFTD model $\mathcal{X}_{t+1} = \left[\!\left[ \{f_{\Theta_n}([I_n^{t+1}]; t+1)\}_{n=1}^3 \right]\!\right] \in \mathbb{R}^{I_1^{t+1} \times I_2^{t+1} \times I_3^{t+1}}$ (Take an example $I_n^{t+1} = I_n^t + 1$) and the old OFTD model $\mathcal{X}_t$ holds with probability at least $1 - \delta$:*

$$|(\mathcal{X}_{t+1})_{(i_1, i_2, i_3)} - (\mathcal{X}_t)_{(i_1, i_2, i_3)}| \leq C_2 \max_n (I_n^t + 1 - i_n), \tag{11}$$

*where $C_2 = 6(\eta^{3d}\omega_0^{3d-3} + \kappa\eta^{2d}\omega_0^{2d-2})\max(I_1^t I_2^t, I_1^t I_3^t, I_2^t I_3^t)$ and $\eta = 2wr^2 + w\ln\frac{3d}{\delta}$.*

*Proof.* First we derive the regret bound for each factor INR $f_{\Theta_n}(\cdot) : \mathbb{R} \to \mathbb{R}^r$. For any $x \in [0, I_n^t]$ we have

$$\|f_{\Theta_n}(x; t+1) - f_{\Theta_n}(x; t)\|_{\ell_1} \leq \left\| f_{\Theta_n}(x; t+1) - \int_{I_n^t}^{I_n^t+1} f_{\Theta_n}(s; t+1)ds \right\|_{\ell_1}$$

$$+ \left\| \int_{I_n^t}^{I_n^t+1} f_{\Theta_n}(s; t+1)ds - \int_{I_n^t}^{I_n^t+1} f_{\Theta_n}(s; t)ds \right\|_{\ell_1} \tag{12}$$

$$+ \left\| \int_{I_n^t}^{I_n^t+1} f_{\Theta_n}(s; t)ds - f_{\Theta_n}(x; t) \right\|_{\ell_1}.$$

From Lemma 1 and its proof, we have that with probability at least $1 - \delta$ the first term on the right-hand side of (12) admits

$$\left\| f_{\Theta_n}(x; t+1) - \int_{I_n^t}^{I_n^t+1} f_{\Theta_n}(s; t+1)ds \right\|_{\ell_1} = \left\| \int_{I_n^t}^{I_n^t+1} f_{\Theta_n}(x; t+1) - f_{\Theta_n}(s; t+1)ds \right\|_{\ell_1}$$

$$\leq \left\| \int_{I_n^t}^{I_n^t+1} \eta^d \omega_0^{d-1}(s - x)ds \right\|_{\ell_1}$$

$$\leq \left\| \int_{I_n^t}^{I_n^t+1} \eta^d \omega_0^{d-1}(I_n^t + 1 - x)ds \right\|_{\ell_1}$$

$$= \eta^d \omega_0^{d-1} |I_n^t + 1 - x|, \tag{13}$$

where $\eta = 2wr^2 + w\ln\frac{3d}{\delta}$. Similarly we can get

$$\left\| \int_{I_n^t}^{I_n^t+1} f_{\Theta_n}(s; t)ds - f_{\Theta_n}(x; t) \right\|_{\ell_1} \leq \eta^d \omega_0^{d-1} |I_n^t + 1 - x|. \tag{14}$$

Define $h_n(x) = f_{\Theta_n}(x; t+1) - f_{\Theta_n}(x; t)$, and from assumptions we have $h_n(I_n^t) = 0$. From the mean value theorem for integral, the second term on the right-hand side of (12) admits

$$\left\| \int_{I_n^t}^{I_n^t+1} f_{\Theta_n}(s; t+1)ds - \int_{I_n^t}^{I_n^t+1} f_{\Theta_n}(s; t)ds \right\|_{\ell_1} = \left\| \int_{I_n^t}^{I_n^t+1} h_n(s)ds \right\|_{\ell_1} = \|h_n(\zeta)\|_{\ell_1}, \tag{15}$$

---

[4]This assumption is reasonable since data streams in real-world often exhibit local continuity and hence the learned OFTD models at adjacent time points would be invariant at the boundary $I_n^t$.

where $\zeta \in (I_n^t, I_n^t + 1)$. By the Lagrange mean value theorem we have

$$\|h_n(\zeta)\|_{\ell_1} = \left\|h_n(\zeta) - h_n(I_n^t)\right\|_{\ell_1} = \left\|h_n^{'}(\xi)(\zeta - I_n^t)\right\|_{\ell_1} \leq 2\kappa|I_n^t + 1 - x|, \qquad (16)$$

where $\xi \in (I_n^t, \zeta)$. Here, we have used the assumption that $\|h_n^{'}(\xi)\|_{\ell_1} = \|f_{\Theta_n}^{'}(\xi; t + 1) - f_{\Theta_n}^{'}(\xi; t)\|_{\ell_1} \leq 2\kappa$. Combining all the results, we obtain:

$$\|f_{\Theta_n}(x; t + 1) - f_{\Theta_n}(x; t)\|_{\ell_1} \leq 2(\eta^d \omega_0^{d-1} + \kappa)|I_n^t + 1 - x|. \qquad (17)$$

Now we can derive the regret bound for online FTD. Write each element of the tensor $\mathcal{X}_t$ as $(\mathcal{X}_t)_{(i_1,i_2,i_3)} = \sum_{m=1}^r f_{\Theta_1}([I_1^t]; t)_{(i_1,m)} f_{\Theta_2}([I_2^t]; t)_{(i_2,m)} f_{\Theta_3}([I_3^t]; t)_{(i_3,m)}$. Then for any $(i_1, i_2, i_3)$ $(i_n \leq I_n^t)$ we have

$$
\begin{aligned}
&\left|(\mathcal{X}_{t+1})_{(i_1,i_2,i_3)} - (\mathcal{X}_t)_{(i_1,i_2,i_3)}\right| \\
&= \left|\sum_{m=1}^r f_{\Theta_1}([I_1^t]; t+1)_{(i_1,m)} f_{\Theta_2}([I_2^t]; t+1)_{(i_2,m)} f_{\Theta_3}([I_3^t]; t+1)_{(i_3,m)}\right.\\
&\quad \left.- f_{\Theta_1}([I_1^t]; t)_{(i_1,m)} f_{\Theta_2}([I_2^t]; t)_{(i_2,m)} f_{\Theta_3}([I_3^t]; t)_{(i_3,m)}\right| \\
&= \left|\sum_{m=1}^r (f_{\Theta_1}([I_1^t]; t+1)_{(i_1,m)} - f_{\Theta_1}([I_1^t]; t)_{(i_1,m)})\ f_{\Theta_2}([I_2^t]; t)_{(i_2,m)} f_{\Theta_3}([I_3^t]; t)_{(i_3,m)}\right.\\
&\quad + (f_{\Theta_2}([I_2^t]; t+1)_{(i_2,m)} - f_{\Theta_2}([I_2^t]; t)_{(i_2,m)})\ f_{\Theta_1}([I_1^t]; t+1)_{(i_1,m)} f_{\Theta_3}([I_3^t]; t)_{(i_3,m)} \\
&\quad \left.+ (f_{\Theta_3}([I_3^t]; t+1)_{(i_3,m)} - f_{\Theta_3}([I_3^t]; t)_{(i_3,m)})\ f_{\Theta_1}([I_1^t]; t+1)_{(i_1,m)} f_{\Theta_2}([I_2^t]; t+1)_{(i_2,m)}\right| \\
&\leq \|f_{\Theta_1}(i_1; t+1) - f_{\Theta_1}(i_1; t)\|_{\ell_1} \|f_{\Theta_2}(i_2; t)\|_{\ell_1} \|f_{\Theta_3}(i_3; t)\|_{\ell_1} \\
&\quad + \|f_{\Theta_2}(i_2; t+1) - f_{\Theta_2}(i_2; t)\|_{\ell_1} \|f_{\Theta_1}(i_1; t+1)\|_{\ell_1} \|f_{\Theta_3}(i_3; t)\|_{\ell_1} \\
&\quad + \|f_{\Theta_3}(i_3; t+1) - f_{\Theta_3}(i_3; t)\|_{\ell_1} \|f_{\Theta_1}(i_1; t+1)\|_{\ell_1} \|f_{\Theta_2}(i_2; t+1)\|_{\ell_1}.
\end{aligned}
$$
$$(18)$$

Substituting (17) into (18) we obtain:

$$
\begin{aligned}
\left|(\mathcal{X}_{t+1})_{(i_1,i_2,i_3)} - (\mathcal{X}_t)_{(i_1,i_2,i_3)}\right| &\leq 2(\eta^d \omega_0^{d-1} + \kappa)|I_1^t + 1 - i_1|(\eta^d \omega_0^{d-1})^2 I_2^t I_3^t \\
&\quad + 2(\eta^d \omega_0^{d-1} + \kappa)|I_2^t + 1 - i_2|(\eta^d \omega_0^{d-1})^2 I_1^t I_3^t \\
&\quad + 2(\eta^d \omega_0^{d-1} + \kappa)|I_3^t + 1 - i_3|(\eta^d \omega_0^{d-1})^2 I_1^t I_2^t \\
&\leq C_2 \max_n(I_n^t + 1 - i_n),
\end{aligned}
$$
$$(19)$$

where $C_2 = 6(\eta^{3d} \omega_0^{3d-3} + \kappa \eta^{2d} \omega_0^{2d-2}) \max(I_1^t I_2^t, I_1^t I_3^t, I_2^t I_3^t)$. The proof is completed. $\qquad \square$

## C  Implementation Details

### C.1  Dataset Details

We introduce details of the used datasets here. For single-aspect streaming, Air Quality[5] contains hourly measurements of pollutant concentrations at different monitoring stations. Indoor Condition[6] includes measurements of humidity and temperature across various indoor locations over time. Radar Traffic[7] records traffic flow data over time at different locations. Chicago Taxi[8] represents taxi trip data in Chicago. Intel Lab Sensor[9] comprises measurements of humidity, temperature, light, and voltage collected by sensors in a laboratory. For multi-aspect streaming, video datasets foreman and carphone[10] and the business review dataset YELP[11] are used. The detailed information is shown in Table 8.

---

[5]https://archive.ics.uci.edu/dataset/501/beijing+multi+site+air+quality+data,    https://www.kaggle.com/datasets/decide-soluciones/air-quality-madrid

[6]https://archive.ics.uci.edu/dataset/374/appliances+energy+prediction

[7]https://www.kaggle.com/datasets/vinayshanbhag/radar-traffic-data

[8]https://data.cityofchicago.org/Transportation/Taxi-Trips-2013-2023-/wrvz-psew

[9]https://db.csail.mit.edu/labdata/labdata.html

[10]http://trace.eas.asu.edu/yuv/

[11]https://www.yelp.com/dataset

Table 8: Real-world tensor datasets used in experiments.

| Name | Modes | Tensor Size |
|---|---|---|
| Beijing Air Quality | locations × pollutants × time | $12 \times 6 \times \mathbf{5994}$ |
| Madrid Air Quality | locations × pollutants × time | $26 \times 17 \times \mathbf{3043}$ |
| Indoor Condition | locations × sensors × time | $9 \times 2 \times \mathbf{2623}$ |
| Radar Traffic | locations × directions × time | $17 \times 5 \times \mathbf{6419}$ |
| Chicago Taxi | sources × destinations × time | $77 \times 77 \times \mathbf{2904}$ |
| Intel Lab Sensor | locations × sensors × time | $52 \times 4 \times \mathbf{1152}$ |
| Video foreman | height × width × frames | $144 \times 176 \times \mathbf{100}$ |
| Video carphone | height × width × frames | $144 \times 176 \times \mathbf{100}$ |
| YELP | user × business × year-month | $1000 \times 992 \times \mathbf{93}$ |

## C.2 Experimental Settings

In experiments, the rank parameter $r$ is set to 100 for all datasets. And the width of INR networks is set to 128 for all datasets. The sine activation $\sin(\omega_0 \cdot)$ is used as nonlinear activation with Lipschitz continuous property. For single-aspect streaming, $\omega_0$ is set to $1.5$ for madrid and sensor, and $0.3$ for other datasets. For multi-aspect streaming, $\omega_0$ is set to $1.5$ for foreman and carphone, and $0.01$ for YELP due to its intense sparsity. The Beta distribution parameter $\alpha$ is set to 1, and $\beta$ is set to 1.2. A proportion of $1/3$ historical data is randomly selected according to the Beta distribution to construct the memory buffer $\mathcal{M}_t$ at each time point $t$. The learning rate of the Adam optimizer is set to 0.001. We perform 100 iterations for single-aspect streaming and 500 iterations for multi-aspect streaming at each time point, using the Adam optimizer. For baseline methods, we tuned their hyperparameters using grid search or following their authors' suggestions to obtain their best performances. For single-aspect streaming, we set the temporal dimension of the data stream to grow by 1 at each time point. For multi-aspect streaming, we set each dimension of the data stream to grow by $10\%$ w.r.t. the total length of the data at each time point. All the experiments are conducted on a computer with i5-12600KF CPU, RTX 4070 Ti SUPER GPU, and 64 GB memories. The Pytorch framework and MATLAB 2024a are used to conduct experiments.

## D   More Results

In this section, we present three main components: additional ablation study results, further discussion of the proposed theory, and several attempts to improve the proposed method.

We provide ablation study of the usage of INR (i.e., OFTD with or without INR) in Fig. 5 and Fig. 6 by using the radar traffic dataset. The use of INR helps to preserve the spatial continuity of the completion results, thus enhancing performances for OFTD (Fig. 5). What's more, OFTD w/ INR better preserves the smoothness of the temporal curve, highlighting the contribution of the INR to the robustness and accuracy of the completion results (Fig. 6).

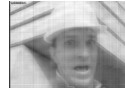 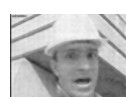 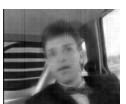 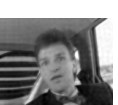

| OFTD w/o INR | OFTD w/ INR | OFTD w/o INR | OFTD w/ INR |
|---|---|---|---|
| NRE 0.205 | NRE 0.084 | NRE 0.226 | NRE 0.111 |

Figure 5: Ablation study for the proposed method without (w/o) and with (w/) the INR on video datasets **foreman** and **carphone**. OFTD w/o INR degrades to the classical online CP decomposition.

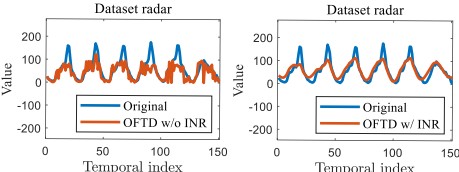

Figure 6: Temporal curves of a spatial pixel reconstructed by OFTD without or with INR under SR 0.3.

Additional ablation study regarding the CP-rank $r$ is shown in Table 9, which shows that our method is quite robust w.r.t. the CP-rank $r$. We also tested the influence of the complexity of the INR (i.e.,

width and depth), as shown in the Table 10 and Table 11. Our OFTD is relatively robust to these parameters. In experiments, we have consistently set the width to **128** and the depth to **3**, which suffice to achieve satisfactory performance across datasets, while further fine-tuning these parameters could achieve even better performance.

Table 9: Ablation study for the CP-rank $r$ in the OFTD model.

| Dataset | Beijing Madrid (average) | | | | | | | | |
|---|---|---|---|---|---|---|---|---|---|
| Rank $r$ | 20 | 40 | 60 | 80 | 100 | 120 | 140 | 160 | 180 |
| NRE | 0.1253 | 0.1255 | 0.1243 | 0.1242 | 0.1245 | 0.1251 | 0.1235 | 0.1246 | 0.1252 |

Table 10: Ablation study for the depth (number of layers) of INRs on the **Condition** data. We vary each factor while fixing the other ones to assess its individual effects.

| Depth | 2 | 3 | 4 | 5 | 6 | 7 | 8 | 9 | 10 |
|---|---|---|---|---|---|---|---|---|---|
| NRE | 0.099 | 0.093 | 0.088 | 0.086 | 0.085 | 0.088 | 0.089 | 0.096 | 0.109 |

Table 11: Ablation study for the width of INRs on the **Condition** data. We vary each factor while fixing the other ones to assess its individual effects.

| Width | 32 | 64 | 128 | 192 | 256 | 320 | 384 | 448 | 512 |
|---|---|---|---|---|---|---|---|---|---|
| NRE | 0.128 | 0.106 | 0.093 | 0.085 | 0.087 | 0.082 | 0.083 | 0.081 | 0.085 |

Also, we have tested our method with ReLU, WIRE Saragadam et al. [2023], FINER Liu et al. [2024], and Sine Sitzmann et al. [2020] activation functions, and the results in the Table 12 show that the default Sine activation function is effective and suitable for our OFTD method. The higher NRE with ReLU activation highlights the importance of activation functions with suitable spectral properties for coordinate-based INRs Sitzmann et al. [2020], like the ones used in our method.

Table 12: Ablation study for the activation functions used in the INRs on the **Condition** data.

| Activation function | WIRE | FINER | ReLU | Sine |
|---|---|---|---|---|
| NRE | 0.105 | 0.103 | 0.215 | 0.093 |

To further demonstrate the applicability of our OFTD on large-scale (e.g., higher-order) tensor datasets, we consider testing on a fourth-order color video dataset of size $3 \times 144 \times 176 \times 100$. Our method can readily extend to the higher-order case by using higher-order CP decomposition parameterized by INRs. The OFTD outperforms the baseline GOCPT for the fourth-order streaming tensor completion: NRE $0.187$ vs. $0.235$ (OFTD vs. GOCPT) using single-aspect settings, demonstrating the effectiveness of OFTD on higher-order tensors. We also provide more experimental results in Fig. 7. Our OFTD achieves generally better performances by observing the more accurate temporal curves.

Regarding the theoretical Lemma 1 and Theorem 1 proposed in the main text, we have the following discussion. First, we qualitatively prove the empirical validity of Lemma 1. We provide more experimental results about the smoothness of the OFTD model w.r.t. parameters $\omega_0$ (the sine activation function parameter) and $w^2$ (the variance of the initialization of weight matrices of the MLP) in Table 13 and 14.

We define a smooth metric $S(\mathcal{X}) := \sum_k \left\| \mathcal{X}_{(:,:,k+1)} - \mathcal{X}_{(:,:,k)} \right\|_{\ell_1}$ to evaluate the smoothness of the recovered tensor. It can be observed that the smoothness of the model is enhanced (i.e., the smooth metric $S$ decreases) with smaller $\omega_0$ and $w$, which coincides with Lemma 1 that smaller $\omega_0$ and $w$ lead to lower Lipschitz smooth bound (Tables 13 and 14). And our method could benefit from an appropriate degree of smoothness brought by tuning these hyperparameters.

Table 13: Ablation study for the hyperparameter $\omega_0$ in the sine activation function $\sin(\omega_0 \cdot)$ used in the INRs of the OFTD model.

| Dataset | Beijing Madrid (average) | | | | | | | | |
|---|---|---|---|---|---|---|---|---|---|
| $\omega_0$ | 0.1 | 0.3 | 0.5 | 0.7 | 0.9 | 1.1 | 1.3 | 1.5 | 1.7 |
| $S(\times 10^4)$ | 9.2 | 12.6 | 13.7 | 13.8 | 15.9 | 21.7 | 23.4 | 24.3 | 25.0 |
| NRE | 0.1410 | 0.1319 | 0.1184 | 0.1168 | 0.1196 | 0.1272 | 0.1358 | 0.1430 | 0.1478 |

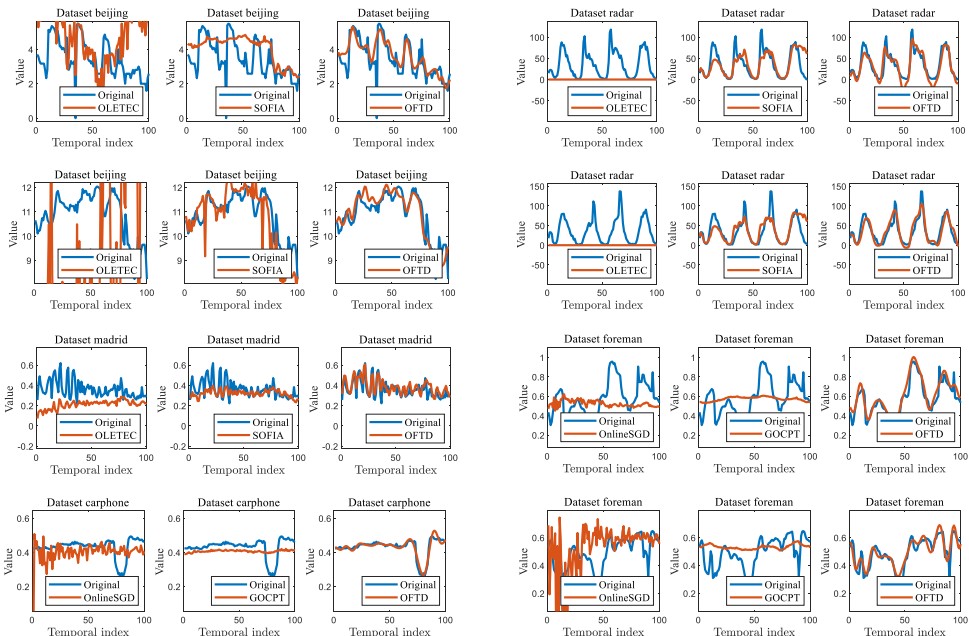

Figure 7: Temporal curves of a spatial pixel reconstructed by different methods for different datasets. The proposed OFTD could better recover the complex nonlinear temporal curves, showcasing its higher accuracy for the streaming data completion problem.

Table 14: Ablation study for the variance $w^2$ of the initialization of weight matrices in the INRs.

| Dataset | Beijing Madrid (average) | | | | | | | | |
|---------|------|------|------|------|------|------|------|------|------|
| $w^2$ | 0.01 | 0.02 | 0.03 | 0.04 | 0.05 | 0.06 | 0.07 | 0.08 | 0.09 |
| $S(\times 10^4)$ | 13.3 | 14.9 | 15.8 | 16.2 | 16.5 | 16.9 | 17.6 | 18.4 | 18.6 |
| NRE | 0.1245 | 0.1280 | 0.1305 | 0.1327 | 0.1337 | 0.1344 | 0.1367 | 0.1405 | 0.1416 |

Then we have the following explanation regarding the assumption of invariance in Theorem 1. The invariant assumption was deduced for general scenarios, i.e., real-world data streams often gradually and smoothly change across the evolving directions, and hence the learned OFTD models at adjacent time points would be invariant at the boundary. In this case, nearby points will not be negatively affected by the new data since they are similar in structure.

To empirically validate the invariant assumption, we calculate the relative error between boundary variables $\frac{\left\|f_n\left(I_n^{t,t+1}\right) - f_n\left(I_n^{t,t}\right)\right\|_2}{\left\|f_n\left(I_n^{t,t}\right)\right\|_2}$ before and after the online optimization at time $t+1$. In the Table 15, we see that the relative error is less than 5%, validating the rationality of the invariant assumption. We note that even without the invariant assumption, the theoretical regret bound and qualitative conclusion of Theorem 1 could still be deduced analogously by adding a small constant $\epsilon$ that indicates the change of boundary variables.

Table 15: Relative error between the boundary variables before and after the online optimization at $t+1$.

| $t+1$ | 20 | 40 | 60 | 80 |
|-------|------|------|------|------|
| Foreman | 4.10% | 2.23% | 3.99% | 1.19% |
| Condition | 1.70% | 4.25% | 3.40% | 1.99% |

Finally, to empirically validate Theorem 1, we test OFTD without memory buffer and report the NRE at different positions after the online optimization at $I_n^t = 100$, as shown in the Table 16. More distant positions (i.e., smaller $i_n$) tend to hold larger NRE without memory buffer, indicating that distant information is more likely to be forgotten. This result precisely aligns with Theorem 1, which states that more distant information holds larger regret bound. Hence, the theoretical result in Theorem 1 is empirically satisfied in practice.

Table 16: NRE results at different positions $i_n$ with and without memory buffer.

| Position $i_n$ | 10 | 20 | 30 | 40 | 50 | 60 | 70 | 80 | 90 |
|---|---|---|---|---|---|---|---|---|---|
| Without memory buffer | 0.440 | 0.314 | 0.300 | 0.275 | 0.256 | 0.228 | 0.170 | 0.127 | 0.101 |
| With memory buffer | 0.075 | 0.104 | 0.074 | 0.071 | 0.078 | 0.098 | 0.077 | 0.100 | 0.093 |

Next, we will present some interesting attempts to improve our methods. According to the definition of $\mathcal{Y}_t$ in Definition (3), the tensor can be extremely sparse. Thus We further conducted experiments by imposing an $l_1$-norm sparse regularization on the factors of the CP decomposition to incorporate the sparse prior: $\lambda \sum_{n=1}^{N} ||f_n([I_t])||_{l_1}$, where $\lambda$ is a sparse regularization parameter. The results with different values are shown in the Table 17. The results show that the sparse regularization with a suitable has the potential to improve the performance of our method, while higher may over-constrain the model. It would be interesting to develop other effective sparse regularizations under the OFTD framework, and we leave it to future research.

Table 17: NRE results with different intensities of the sparse regularization.

| $\lambda$ | 0 | 0.01 | 0.03 | 0.05 | 0.07 | 0.09 |
|---|---|---|---|---|---|---|
| Foreman | 0.084 | 0.081 | 0.079 | 0.078 | 0.078 | 0.080 |
| Condition | 0.093 | 0.091 | 0.088 | 0.088 | 0.090 | 0.092 |

Currently the Beta parameter $\beta$ is fixed. It is a great idea to dynamically adjust the Beta parameter based on the learning process. We conduct experiments using a linear dynamic scheme for $\beta$ (i.e., $\beta_t = \beta_0 + t\Delta\beta$, where $\beta_0 = 1$), as shown in Table 18. Better NRE results are observed when $\Delta\beta$ is a positive value (i.e., long-tail distribution). Conversely, when $\Delta\beta$ is a negative number, the gains are negative. We attribute this result to the nature of our forgetting bound, which depends on the relative position $(I_n^t + 1 - i_n)$ (as shown in Eq. (8) of the main text). It shows that distant information is more likely to be forgotten, and hence a long-tail memory buffer with $\Delta\beta > 0$ is favored. The design of more complex $\beta_t$ schedules and their theoretical analysis are not trivial due to the complexity of the optimization problem. We leave this promising direction to future research.

Table 18: NRE results with dynamic scheme $\beta_t = 1 + t\Delta\beta$ on the **Foreman** data.

| $\Delta\beta$ | -0.01 | -0.005 | 0 | 0.005 | 0.01 | 0.015 | 0.02 | Fix $\beta_t = 1.2$ |
|---|---|---|---|---|---|---|---|---|
| NRE | 0.092 | 0.087 | 0.086 | 0.083 | 0.083 | 0.084 | 0.084 | 0.084 |

If we want to further reduce the computational complexity associated with memory buffer, we can consider using fixed-size memory buffer strategies (non-increasing memory buffer size). Especially, we utilize the Beta distribution to construct long-tail memory buffer that contains up to 100 historical samples, which strictly restricts computational complexity.

From the results in the Table 19, we can observe that OFTD still performs stably under the fixed-size memory buffer. In particular, OFTD with fixed-size memory buffer achieves a significantly lower NRE of 0.1311 (vs. 0.3979 without memory buffer) with only a moderate increase in FLOPs (10.79M vs. 9.09M) on the Madrid case. The results demonstrate the practical efficiency and effectiveness of the fixed-size memory buffer strategy. It is also possible to consider other continual learning strategies such as memory buffer-free methods (e.g., gradient projection Lin et al. [2022] and prototype-sampling Asadi et al. [2023]) to further enhance the efficiency of OFTD, yet the combination is not trivial due to the distinct structures of OFTD. Hence, we leave this direction for future research .

Table 19: Ablation study for the memory buffer size (percentage of buffer size w.r.t. the whole tensor size), including a fixed-size memory buffer (up to 100 samples).

| Dataset | Beijing | | | | | | Madrid | | | | | |
|---|---|---|---|---|---|---|---|---|---|---|---|---|
| Buffer size | 100% | 50% | 33% | 20% | Fixed-size | 0% | 100% | 50% | 33% | 20% | Fixed-size | 0% |
| NRE | 0.1280 | 0.1299 | 0.1308 | 0.1324 | 0.1422 | 0.2903 | 0.1112 | 0.1160 | 0.1182 | 0.1204 | 0.1311 | 0.3979 |
| FLOPs | 87.97M | 43.99M | 29.33M | 17.61M | 8.89M | 6.03M | 139.42M | 69.73M | 46.50M | 27.93M | 10.79M | 9.09M |

In addition to the above attempts, the OFTD method can be readily generalized by combining with other tensor decomposition paradigms such as Tucker decomposition and tensor-train decomposition.

Especially, we define the Tucker functional tensor decomposition (Tucker-FTD) as

$$\mathcal{X}_t = \mathcal{C} \times_1 f_1\left([I_1^t]\right) \times_2 f_2\left([I_2^t]\right) \times_3 f_3\left([I_3^t]\right) \in \mathbb{R}^{I_1^t \times I_2^t \times I_3^t},$$

where $\mathcal{C} \in \mathbb{R}^{r_1 \times r_2 \times r_3}$ is a core tensor and $f_n\left([I_n^t]\right) \in \mathbb{R}^{I_n^t \times r_n}$ $(n = 1, 2, 3)$ are factor functions parameterized by INRs.

Moreover, we define the tensor-train functional tensor decomposition (TT-FTD) as

$$\mathcal{X}_t = f_2\left([I_2^t]\right) \times_1 f_1\left([I_1^t]\right) \times_3 f_3\left([I_3^t]\right) \in \mathbb{R}^{I_1^t \times I_2^t \times I_3^t},$$

where $f_2\left([I_2^t]\right) \in \mathbb{R}^{r_1 \times I_2^t \times r_3}$, $f_1\left([I_1^t]\right) \in \mathbb{R}^{I_1^t \times r_1}$, and $f_3\left([I_3^t]\right) \in \mathbb{R}^{r_3 \times I_3^t}$ are factor functions parameterized by INRs.

To adapt the Tucker-FTD and TT-FTD for streaming data, we consider their online versions by progressively expanding the input coordinate vectors $[I_n^t]$ of INRs, and hence expanding the sizes of factor matrices/tensors to fit the new tensor size. We give a numerical example by comparing different tensor decompositions under the OFTD framework, as shown in the Table 20. We control the rank of different methods such that they hold similar number of learnable parameters.

Table 20: NRE results by OFTD with different tensor decomposition paradigms on the **Foreman** data.

| Method | TT | CP | Tucker |
|---|---|---|---|
| NRE | 0.085 | 0.084 | 0.082 |
| Number of Params. | $3.12 \times 10^5$ | $2.76 \times 10^5$ | $2.93 \times 10^5$ |

Tucker-FTD achieves a relatively better result in this case, which can be rationally explained by the additional core tensor parameter $\mathcal{C}$ that enhances the representation ability of Tucker-FTD. In future research, we can consider more tensor decomposition paradigms such as the tensor SVD under the OFTD framework.

