# OpenReview forum: "Online Functional Tensor Decomposition via Continual Learning for Streaming Data Completion"
_NeurIPS.cc/2025/Conference — NeurIPS 2025 spotlight_

### Official Review · Reviewer_beqG · 2025-06-23

**Clarity:** 3
**Significance:** 3
**Originality:** 2
**Rating:** 5
**Confidence:** 4

**Summary:**

This paper proposes a novel framework, Online Functional Tensor Decomposition (OFTD), for streaming tensor data completion. The core idea is to model CP decomposition factors as Implicit Neural Representations (INRs), thus to capture spatial-temporal continuity and
low-dimensional compact structures of streaming data . To mitigate catastrophic forgetting, this work adopts a long-tail memory replay mechanism and theoretically establishes a regret bound. Extensive experiments on real-world single- and multi-aspect streaming datasets show that OFTD outperforms a wide range of baselines in terms of NRE for the most cases.

**Questions:**

1. Why CP decompositions?  Is it possible to generalize this method to other tensor decompositions like TSVD or Tensor Train?

2. How does the regret bound correlate empirically with performance degradation?

**Ethical Concerns:**

["NO or VERY MINOR ethics concerns only"]

**Final Justification:**

My concerns have been fully addressed.

**Limitations:**

yes

**Paper Formatting Concerns:**

No Paper Formatting Concerns

**Quality:**

3

**Strengths And Weaknesses:**

**Strengths:**

1. The integration of INR with CP decomposition for streaming tensor data completion is novel and well-motivated.

2.  The method is validated on both single-aspect and multi-aspect tensor data streams with clear performance improvements over the baselines.

3. Authors provide a regret bound that explains the forgetting behavior of the model.

**Weaknesses:**

1. Although the work is   motivated by continual learning, connections to and comparisons with classic continual learning approaches are missing.

2. The performance of OFTD may be sensitive to the architecture or hyperparameters of the INR (e.g., depth, activation), but this is not explored or ablated in the current version.

---

> ### Author Rebuttal · Authors · 2025-07-25
>
> We sincerely thank the reviewer for the valuable and constructive comments! We have carefully made more discussions and clarifications according to your comments, including more discussions on the connections between our method and classical continual learning paradigms, more ablation studies of the INR architectures, more tensor decomposition paradigms under the OFTD framework, and more empirical validations of the theoretical regret bound analysis. Please see details in the following.
>
> >**W1: Although the work is motivated by continual learning, connections to and comparisons with classic continual learning approaches are missing.**
>
> Thanks for this constructive comment. Here, we further discuss the connections between our method and continual learning in a more comprehensive manner. The continual learning refers to a machine learning paradigm where a model sequentially learns from a non-stationary stream of data over time, while retaining historical knowledge [R1]. The proposed OFTD belongs to the continual learning paradigm, which learns the INR networks from a continuous stream of tensor data, with such information becoming progressively available over time. Hence, the proposed OFTD has the potential to be combined with proven methods in continual learning (e.g., memory replay) to alleviate the forgetting of historical tensor knowledge. To this end, we have proposed a long-tail memory reply method to effectively alleviate forgetting. The combination of our method with more other continual learning methods (e.g., gradient projection [R2] and prototype-sampling [R3]) to alleviate the forgetting is also an interesting research direction, yet the combination is not trivial due to the distinct structures of OFTD. Hence, we leave this direction for future research.
>
> As for the comparisons with classical continual learning methods, we want to clarify that the proposed OFTD is the first method to introduce the continual learning paradigm for the streaming data completion task, hence direct comparison with classical continual learning methods designed for tasks like image classification [R2] is not straightforward or directly applicable. Nevertheless, we can numerically compare the proposed long-tail memory reply continual learning method with the vanilla uniformly distributed memory reply continual learning method under the OFTD framework. The results reveal that the long-tail memory reply is indeed more effective, e.g., NRE 0.1233 vs. 0.1261 (long-tail vs. uniform) (more details are shown in Table 5 of the main text). We will include these discussions in the final paper to enhance the clarity and comprehensiveness according to your comments.
>
> [R1] Continual lifelong learning with neural networks: A review, Neural Networks
>
> [R2] TRGP: Trust region gradient projection for continual learning, ICLR
>
> [R3] Prototype-sample relation distillation: towards replay-free continual learning, ICML
>
> >**W2: The performance of OFTD may be sensitive to the architecture or hyperparameters of the INR (e.g., depth, activation), but this is not explored or ablated in the current version.**
>
> As suggested, we tested the influence of the architectures of INRs (width, depth, and activation functions) on the model performance, as shown in the Table below. Our method is relatively robust to the width and depth parameters. Also, we have tested our method with ReLU, WIRE [R4], FINER [R5], and Sine [R6] activation functions, and the results show that the default Sine activation function is effective and suitable for our OFTD method. The higher NRE with ReLU activation highlights the importance of activation functions with suitable spectral properties for coordinate-based INRs [R6], like the ones used in our method.
>
> Table: Ablation study for the width, depth, and activation function of INRs on the Condition data. We vary each factor while fixing the other ones to assess its individual effects.
> |Width|32|64|128|192|256|320|384|448|512|
> |-|-|-|-|-|-|-|-|-|-|
> |NRE|0.128|0.106|0.093|0.085|0.087|0.082|0.083|0.081|0.085|
>
> |Depth|2|3|4|5|6|7|8|9|10|
> |-|-|-|-|-|-|-|-|-|-|
> |NRE|0.099|0.093|0.088|0.086|0.085|0.088|0.089|0.096|0.109|
>
> |Activation function|WIRE|FINER|ReLU|Sine|
> |-|-|-|-|-|
> |NRE|0.105|0.103|0.215|0.093|
>
> [R4] WIRE: Wavelet Implicit Neural Representations, CVPR
>
> [R5] FINER: Flexible spectral-bias tuning in Implicit Neural Representation by Variable-periodic Activation Functions, CVPR
>
> [R6] Implicit Neural Representations with Periodic Activation Functions, NeurIPS
>
> >**Q1: Why CP decompositions? Is it possible to generalize this method to other tensor decompositions like TSVD or Tensor Train?**
>
> Yes! The OFTD method can be readily generalized by combining with other tensor decomposition paradigms such as Tucker decomposition and tensor-train decomposition. Especially, we define the Tucker functional tensor decomposition (Tucker-FTD) as
> $$
> {\mathcal X}_t ={\mathcal C}\times_1 f_1([I_1^t]) \times_2 f_2([I_2^t]) \times_3 f_3([I_3^t])\in{\mathbb R}^{I_1^t\times I_2^t\times I_3^t},
> $$
> where ${\mathcal C}\in{\mathbb R}^{r_1\times r_2\times r_3}$ is a core tensor and $f_n([I_n^t])\in{\mathbb R}^{ I_n^t \times r_n}$ ($n=1,2,3$) are factor functions parameterized by INRs. Moreover, we define the tensor-train functional tensor decomposition (TT-FTD) as
> $$
> {\mathcal X}_t =f_2([I_2^t]) \times_1f_1([I_1^t]) \times_3 f_3([I_3^t])\in{\mathbb R}^{I_1^t\times I_2^t\times I_3^t},
> $$
> where $f_2([I_2^t])\in{\mathbb R}^{r_1\times I_2^t\times r_3}, f_1([I_1^t])\in{\mathbb R}^{I_1^t\times r_1}, f_3([I_3^t])\in{\mathbb R}^{I_3^t\times r_3}$ are factor functions parameterized by INRs. To adapt the Tucker-FTD and TT-FTD for streaming data, we consider their online versions by progressively expanding the input coordinate vectors $[I_n^t]$ of INRs, and hence expanding the sizes of factor matrices/tensors to fit the new tensor size. We give a numerical example by comparing different tensor decompositions under the OFTD framework, as shown in the Table below. We control the rank of different methods such that they hold similar number of learnable parameters. Tucker-FTD achieves a relatively better result in this case, which can be rationally explained by the additional core tensor parameter $\\mathcal C$ that enhances the representation ability of Tucker-FTD. In future research, we can consider more tensor decomposition paradigms such as the tensor SVD under the OFTD framework. Thanks.
>
> Table: NRE results by OFTD with different tensor decomposition paradigms on the Foreman data.
> |Method|TT|CP|Tucker|
> |-|-|-|-|
> |NRE|0.085|0.084|0.082|
> |Number of Params.|$3.12\\times 10^{5}$|$2.76\\times 10^{5}$|$2.93\\times 10^{5}$|
>
> >**Q2: How does the regret bound correlate empirically with performance degradation?**
>
> Thanks for this insightful comment! The theoretical regret bound analysis in Theorem 1 reveals that more distant information is likely to hold larger regret bound (i.e., more easily to be forgotten) due to the distance term $(I_n^t +1-i_n)$. To empirically validate this theoretical result, we run the OFTD without memory buffer and report the NRE at different positions after the online optimization at $I_n^t=100$. We can see that more distant positions (i.e., smaller $i_n$) tend to hold larger NRE when the memory buffer is disabled, indicating that more distant information is more likely to be forgotten, precisely aligning with the theoretical result. We can also observe that introducing the long-tail memory buffer method could largely alleviate the forgetting, as shown in the Table below.
>
> Table: NRE results at different positions $i_n$ with and without memory buffer on the Foreman data.
> |Position $i_n$|10|20|30|40|50|60|70|80|90|100|
> |-|-|-|-|-|-|-|-|-|-|-|
> |Without memory buffer|0.440|0.314|0.300|0.275|0.256|0.228|0.170|0.127|0.101|0.065|
> |With memory buffer|0.075|0.104|0.074|0.071|0.078|0.098|0.077|0.100|0.093|0.066|
>
> Should you need further information, please let us know. We look forward to hearing from your feedback!

---

> > ### Comment · Reviewer_beqG · 2025-08-02
> >
> > Thank you very much for your response! Your reply addressed most of my concerns. Based on your answers, I now have a follow-up question:  I believe your model should also be applicable to matrix data, would it be possible to reshape the tensor data into a matrix and process it accordingly? I'm curious about how leveraging the tensor structure, as opposed to a matrix representation, affects the model's performance.

---

> > > ### Author Response · Authors · 2025-08-02
> > >
> > > We sincerely thank the reviewer’s constructive comments and recognition of our responses! Regarding your question about the generalization to matrix data, we made further clarifications as follows.
> > >
> > > >**Comment: I believe your model should also be applicable to matrix data, would it be possible to reshape the tensor data into a matrix and process it accordingly? I'm curious about how leveraging the tensor structure, as opposed to a matrix representation, affects the model's performance.**
> > >
> > > Yes! Our method can be readily applied to matrix data by considering the following matrix decomposition model parameterized by INRs:
> > > $$
> > > {\bf X}_t=f_1([I_1^t])f_2([I_2^t])^T\in{\mathbb R}^{I_1^t\times I_2^t},
> > > $$
> > > where $f_1([I_1^t])\in{\mathbb R}^{I_1^t\times r}, f_2([I_2^t])\in{\mathbb R}^{I_2^t\times r}$ are matrix factors parameterized by INRs. We consider its online version by progressively expanding the input coordinate vectors $[I_1^t], [I_2^t]$, and hence expanding the sizes of factor matrices to fit the new data size. To test this matrix representation model, we reshape the size of the video tensor data from $n_1\times n_2\times n_3$ to $n_1n_2\times n_3$ and apply the matrix representation for streaming data completion along the $n_3$ dimension (single-aspect setting). We also run the original tensor representation-based method OFTD under the same setting. We control the rank $r$ of each method to obtain optimal performance. As shown in the Table below, the tensor-based OFTD outperforms the matrix representation. This can be rationally explained by the fact that the matrix representation and the reshaping operator somewhat break the spatial structure of the tensor data. Such spatial structure (e.g., spatial smoothness) can be useful under the tensor-based OFTD framework, which utilizes the tensor decomposition to exploit low-dimensionality across multiple dimensions, and leverages the INRs to capture more comprehensive spatial-temporal correlations. Hence, it is reasonable that the tensor-based OFTD would outperform the matrix representation method for streaming tensor data completion.
> > >
> > > Table: NRE results by the tensor-based OFTD and the matrix representation on the Foreman and Carphone video data under single-aspect setting.
> > > |Method|Matrix representation|OFTD|
> > > |-|-|-|
> > > |Foreman|0.149|0.105|
> > > |Carphone|0.160|0.118|
> > >
> > > Should you need further information, please let us know. Thanks again!

---

> > > > ### Comment · Reviewer_beqG · 2025-08-02
> > > >
> > > > Thank you so much for your response. My concerns have been fully addressed. I will increase my score by 1 point.

---

> > > > > ### Author Response · Authors · 2025-08-03
> > > > >
> > > > > We thank the reviewer again for the constructive comments and questions!

---

### Official Review · Reviewer_kMsk · 2025-06-29

**Clarity:** 3
**Significance:** 3
**Originality:** 3
**Rating:** 4
**Confidence:** 3

**Summary:**

This paper aims to tackle the challenges in high-velocity streaming tensor data analysis, where new data expands across multiple dimensions of the original dataset. Specifically, it leverages implicit neural representations to model the tensor decomposition factors, thereby enhancing the capability of dynamic data structure modeling. To mitigate the forgetting of historical data when fitting new information, the paper adopts a continual learning approach and develops a memory replay mechanism to address this issue. Theoretically, the paper analyzes the spatial-temporal smoothness regularization and the forgetting pattern of the proposed method during the learning process. In the experimental section, the paper demonstrates the effectiveness of the proposed method on real-world datasets.

**Questions:**

How does the complexity of the INR affect the model performance?

**Ethical Concerns:**

["NO or VERY MINOR ethics concerns only"]

**Final Justification:**

The paper is conceptually new; in the rebuttal, the authors address the concern regarding the computational complexity, the model performance, and also the theoretical results by adding additional results and analysis.

**Limitations:**

yes

**Paper Formatting Concerns:**

I have reviewed the formatting and have no concerns.

**Quality:**

2

**Strengths And Weaknesses:**

### **Strengths**:

• The idea of introducing Functional Tensor Decomposition to the online tensor analysis setup is novel. Unlike classical online tensor analysis methods, it explores the nonlinear relationships of the tensor factors, which is a significant enhancement in streaming data analysis.

• The paper demonstrates an interesting combination of concepts between continual learning and streaming tensor analysis, which may inspire future research.

• The paper is technically sound, with sufficient theoretical analysis to support the effectiveness and the motivation behind the proposed method.


### **Weaknesses**:

• The proposed method tends to forget information distant from the new tensor data, leading the paper to propose a memory replay mechanism to avoid this. However, as more and more new data is introduced, the memory buffer must grow larger and larger. This seems like a serious side effect (e.g., increasing computational complexity), especially when real-time processing is a critical consideration in streaming data analysis.

• The theoretical analysis seems to hold only when the weight matrices of INR during training remain i.i.d. $\mathcal{N}(0,\omega^{2})$. How can this be ensured? Additionally, the first assumption in Theorem 1 seems questionable, as it assumes the boundary variables of the model are invariant before and after training on the new data. What if the deep model severely overfits the new data? If this happens, intuitively, even the nearest point to the new data will be affected by this overfitting.

• There are several existing studies on Functional Tensor Decomposition. For example, the method presented in [1] appears to achieve the same objective as INR. The paper misses references to these types of works.

Reference:

[1]. Functional Bayesian Tucker Decomposition for Continuous-indexed Tensor Data, ICLR, 2024

---

> ### Author Rebuttal · Authors · 2025-07-25
>
> We sincerely thank the reviewer for the valuable and constructive comments! We have carefully made more discussions and clarifications according to your comments, including more discussions on the memory buffer, more validations on the theoretical assumptions, more discussions on related work, and more experiments for the influence of INR complexity. Please see details in the following.
>
> >**W1: The proposed method tends to forget information distant from the new tensor data, leading the paper to propose a memory replay mechanism to avoid this. However, as more and more new data is introduced, the memory buffer must grow larger and larger. This seems like a serious side effect (e.g., increasing computational complexity), especially when real-time processing is a critical consideration in streaming data analysis.**
>
> Yes, in our current implementation, the memory buffer grows larger during online optimization. Nevertheless, OFTD is still practically efficient and could achieve real-time processing (less than one second per online optimization). The reason of such efficiency can be illustrated from two aspects. First, OFTD is relatively robust to the memory buffer size and could perform stably even using a small proportion of historical data (e.g., 20%), as shown in the Table below. Thus, OFTD could achieve efficient real-time online processing with such small memory buffer.
>
> Second, if we want to further reduce the computational complexity associated with memory buffer, we can consider using fixed-size memory buffer strategies (non-increasing memory buffer size). Especially, we utilize the Beta distribution to construct long-tail memory buffer that contains up to 100 historical samples, which strictly restricts computational complexity. From the results in the Table below, we can observe that OFTD still performs stably under the fixed-size memory buffer. In particular, OFTD with fixed-size memory buffer achieves a significantly lower NRE of 0.1311 (vs. 0.3979 without memory buffer) with only a moderate increase in FLOPs (10.79M vs. 9.09M) on the Madrid case. The results demonstrate the practical efficiency and effectiveness of the fixed-size memory buffer strategy. It is also possible to consider other continual learning strategies such as memory buffer-free methods (e.g., gradient projection [R1] and prototype-sampling [R2]) to further enhance the efficiency of OFTD, yet the combination is not trivial due to the distinct structures of OFTD. Hence, we leave this direction for future research.
>
> Table: Ablation study for the memory buffer size (the proportion of buffer size w.r.t. the
> whole tensor size), and the fixed-size memory buffer with up to 100 historical samples. The Beijing data contains 5994 temporal samples while Madrid contains 3043 temporal samples.
> |Dataset|Beijing||||||Madrid||||||
> |-|-|-|-|-|-|-|-|-|-|-|-|-|
> |Buffer size|100%|50%|33%|20%|Fixed-size|0%|100%|50%|33%|20%|Fixed-size|0%|
> |NRE|0.1280|0.1299|0.1308|0.1324|0.1422|0.2903|0.1112|0.1160|0.1182|0.1204|0.1311|0.3979|
> |FLOPs|87.97M|43.99M|29.33M|17.61M|8.89M|6.03M|139.42M|69.73M|46.50M|27.93M|10.79M|9.09M|
>
> [R1] TRGP: Trust region gradient projection for continual learning, ICLR
>
> [R2] Prototype-sample relation distillation: towards replay-free continual learning, ICML
>
> >**W2: The theoretical analysis seems to hold only when the weight matrices of INR during training remain i.i.d. ${\mathcal N}(0,w^2)$. How can this be ensured? Additionally, the first assumption in Theorem 1 seems questionable, as it assumes the boundary variables of the model are invariant before and after training on the new data. What if the deep model severely overfits the new data? If this happens, intuitively, even the nearest point to the new data will be affected by this overfitting.**
>
> Thanks! Regarding the Gaussian assumption, we want to clarify that the i.i.d. ${\cal N}(0,w^2)$ for neural network parameters is a widely used assumption in theoretical studies of neural networks [R3,R4]. We follow this assumption to simplify the theoretical proof, while other statistical distributions (e.g., uniform distributions, Poisson distribution, and not necessarily i.i.d.) could also fit in our theoretical results by slightly changing the Lipschitz bound $C_1$ in Lemma 1. Moreover, the empirical validity of Lemma 1 has been qualitatively justified in Tables 10-11 in the Appendix, where we vary the values of $\omega_0$ and $w$ in the Lipschitz bound and observe the empirical smoothness change of the recovered tensor. The smoothness changing trend of the recovered tensor precisely aligns with the changing trend of the Lipschitz bound w.r.t. the parameters $\omega_0$ and $w$ deduced in Lemma 1. Hence, the theoretical result in Lemma 1 is empirically satisfied in practice.
>
> Regarding the invariant assumption, we want to clarify from the following aspects:
>
> 1) First, the OFTD model has a certain degree of robustness against outliers due to the regularization effect induced by the tensor decomposition and the inherent smoothness of INRs, which can alleviate overfitting and ensure stability.
>
> 2) Second, the invariant assumption was deduced for general scenarios, i.e., real-world data streams often gradually and smoothly change across the evolving directions, and hence the
> learned OFTD models at adjacent time points would be invariant at the boundary. In this case, nearby points will not be negatively affected by the new data since they are similar in structure. To empirically validate the invariant assumption, we calculate the relative error between boundary variables $\frac{\lVert f_n(I_n^{t};t+1)-f_n(I_n^{t};t)\rVert _{\ell_2}}{\lVert f_n(I_n^{t};t)\rVert _{\ell_2}}$ before and after the online optimization at time $t+1$. We see that the relative error is less than 5%, validating the rationality of the invariant assumption. We note that even without the invariant assumption, the theoretical regret bound and qualitative conclusion of Theorem 1 could still be deduced analogously by adding a small constant $\epsilon$ that indicates the change of boundary variables.
>
> Table: Relative error between the boundary variables before and after the online optimization at $t+1$.
> |$t+1$|20|40|60|80|
> |-|-|-|-|-|
> |Foreman|4.10%|2.23%|3.99%|1.19%|
> |Condition|1.70%|4.25%|3.40%|1.99%|
>
> 3) Third, if severe overfitting on new data occurs, the boundary invariant assumption may be violated. This could happen for abruptly changed data streams where the data smoothness disappears. To alleviate the ill-posedness under this scenario, we have further proposed a temporal online affine regularizer (Section A.3 in Appendix) to relax the data prior assumption in this extreme case. This makes OFTD more robust and stable in diverse scenarios.
>
> 4) Finally, to empirically validate Theorem 1, we test OFTD without memory buffer and report the NRE at different positions after the online optimization at $I_n^t=100$, as shown in the Table below. More distant positions (i.e., smaller $i_n$) tend to hold larger NRE without memory buffer, indicating that distant information is more likely to be forgotten. This result precisely aligns with Theorem 1, which states that more distant information holds larger regret bound. Hence, the theoretical result in Theorem 1 is empirically satisfied in practice.
>
> Table: NRE results at different positions $i_n$ with and without memory buffer on the Foreman data.
> |Position $i_n$|10|20|30|40|50|60|70|80|90|
> |-|-|-|-|-|-|-|-|-|-|
> |Without memory buffer|0.440|0.314|0.300|0.275|0.256|0.228|0.170|0.127|0.101|
> |With memory buffer|0.075|0.104|0.074|0.071|0.078|0.098|0.077|0.100|0.093|
>
> [R3] Neural Tangent Kernel: Convergence and Generalization in Neural Networks, NeurIPS
>
> [R4] Fine-Grained Analysis of Optimization and Generalization for Overparameterized Two-Layer Neural Networks, ICML
>
> >**W3: There are several existing studies on Functional Tensor Decomposition. For example, the method presented in [1] appears to achieve the same objective as INR. The paper misses references to these types of works.**
>
> Thanks for bringing this related work to our attention! Fang et al. [R5] introduced the functional Bayesian Tucker decomposition, which utilizes Gaussian processes as functional priors to model the factors of the Tucker decomposition, with applications for tensor completion. Compared to this work, we propose the online functional tensor decomposition, which utilizes implicit neural representations (INRs) parameterized by deep neural networks to model tensor factors for streaming data completion. The INRs hold stronger representation ability than the statistical Gaussian processes for characterizing real-world tensor data. Moreover, the proposed OFTD is capable of handling streaming tensor data in an online manner, which is different from the batch tensor decomposition technique in [R5]. We will introduce and discuss this related work in the main text.
>
> [R5] Functional Bayesian Tucker Decomposition for Continuous-indexed Tensor Data, ICLR
>
> >**Q1: How does the complexity of the INR affect the model performance?**
>
> As suggested, we tested the influence of the complexity of the INR (i.e., width and depth), as shown in the Table below. Our OFTD is relatively robust to these parameters. In experiments, we have consistently set the width to $128$ and the depth to $3$, which suffice to achieve satisfactory performance across datasets, while further fine-tuning these parameters could achieve even better performance.
>
> Table: Ablation study for the width and depth of INRs on the Condition data. We vary each factor while fixing the other one to assess its individual effects.
> |Width|32|64|128|192|256|320|384|448|512|
> |-|-|-|-|-|-|-|-|-|-|
> |NRE|0.128|0.106|0.093|0.085|0.087|0.082|0.083|0.081|0.085|
>
> |Depth|2|3|4|5|6|7|8|9|10|
> |-|-|-|-|-|-|-|-|-|-|
> |NRE|0.099|0.093|0.088|0.086|0.085|0.088|0.089|0.096|0.109|
>
> Should you need further information, please let us know. We look forward to hearing from your feedback!

---

> > ### Comment · Reviewer_kMsk · 2025-08-02
> >
> > I would like to thank the authors for addressing most of my concerns, and as a result, I am revising my evaluation and increasing the grade. The additional results are very helpful, and I hope the authors can include this new analysis in the updated manuscript.

---

> > > ### Author Response · Authors · 2025-08-02
> > >
> > > We sincerely thank the reviewer’s constructive comments and recognition of our responses! We will include the new analysis in the updated manuscript as suggested by the reviewer. Thanks again!

---

### Official Review · Reviewer_ZeJq · 2025-07-03

**Clarity:** 4
**Significance:** 4
**Originality:** 3
**Rating:** 5
**Confidence:** 4

**Summary:**

The work proposes an online tensor decomposition framework that incorporates implicit neural representation (INR) strategy that helps with more dynamic continual learning. The tensor decomposition is performed in an online manner at every time step with limited forgetting of the previous tensor knowledge. Towards this, a long-tail memory replay method by constructing a memory buffer controlled by a long-tail beta distribution is introduced. Experiments are presented to evaluate the proposed approach and compare with the baselines.

**Questions:**

1.	In Eq. (2) while introducing the optimization at time step t, the temporal relation is not specified for the factor matrices.

2.	According to the definition of ${\cal Y}_t$ in Definition 3, the tensor can be extremely sparse. Yet, such prior knowledge is not employed in (6), which could limit the performance of the approach.

3.	Minor comments: There is some formatting issue in terms of using the citations to the references. The author names are repeated twice which is better to avoid.

4.	According to Theorem1, not only the distance from new data stream, but also the neural network architecture and the choice of $r$ also plays role in forgetting. There is no discussion regarding that to give some insights in the practical settings.

5.	The parameters of the beta distribution is currently fixed, but according to the theorem, it is dependent on the learning process and can be influenced by that.

6.	There are other influencing factors such as rank, the choice of omega, the number of layers. How are these factors affecting the performance and be selected for optimal outputs.

7.	I could not see much information presented regarding the baselines and their tensor decomposition methods. Are all of them CP? Or have you tried with other tensor decomposition methods such as Tucker etc?

**Ethical Concerns:**

["NO or VERY MINOR ethics concerns only"]

**Final Justification:**

The work advances online tensor data analysis and have proposed a strong benchmark for the future work. The responses to my concerns have been addressed well in the rebuttal phase. Hence, I am happy to raise the point.

**Quality:**

3

**Strengths And Weaknesses:**

Strengths:

1. The paper is well-written and hence it is easy to follow. The notations are clearly defined, and theorems are well presented.

2. INR has emerged as a powerful technique in computer vision and has not much explored in online tensor decomposition. Hence, the work can spark some follow-up studies in this domain utilizing these powerful tools.

Weaknesses:

1. Experimental study presented in the main paper is limited. It looks like the completion is performed in small scale datasets. The dataset size and details are not clearly articulated. More ablation studies are also expected to better evaluate the approach.

---

> ### Author Rebuttal · Authors · 2025-07-26
>
> We sincerely thank the reviewer for the valuable and constructive comments! We have carefully made more discussions and clarifications according to your comments, including more dataset details, more ablation studies, numerical tests with sparse priors, more discussions on Theorem 1 and the Beta distribution. Please see details in the following.
>
> >**W1: Experimental study presented in the main paper is limited. It looks like the completion is performed in small scale datasets. The dataset size and details are not clearly articulated. More ablation studies are also expected to better evaluate the approach.**
>
> Thanks! The dataset sizes and details are articulated in Section C.1 and Table 8 in the Appendix after the main text (in the same PDF file). All datasets are publicly available, and the detailed websites for downloading these datasets are provided in the footnotes under Section C.1. Some datasets are large-scale tensors, e.g., the dataset YELP has a size of 1000 × 992 × 93. To further demonstrate the applicability of our OFTD on large-scale (e.g., higher-order) tensor datasets, we consider testing on a fourth-order color video dataset of size 3×144×176×100. Our method can readily extend to the higher-order case by using higher-order CP decomposition parameterized by INRs. The OFTD outperforms the baseline GOCPT for the fourth-order streaming tensor completion: NRE 0.187 vs. 0.235 (OFTD vs. GOCPT) using single-aspect settings, demonstrating the effectiveness of OFTD on higher-order tensors. In terms of ablation study, please refer to the response to your question Q6. Thanks.
>
> >**Q1: In Eq. (2) while introducing the optimization at time step $t$, the temporal relation is not specified for the factor matrices.**
>
> Thanks! We will revise this equation and introduce the temporal relation in the factor matrices by denoting them as $\\{{\bf U}^{(n)}_t\\}$.
>
> >**Q2: According to the definition of ${\mathcal Y}_t$ in Definition 3, the tensor can be extremely sparse. Yet, such prior knowledge is not employed in (6), which could limit the performance of the approach.**
>
> It is a great idea to further incorporate the sparse prior into the model. Following your comments, we further conducted experiments by imposing an $\ell_1$-norm sparse regularization on the factors of the CP decomposition to incorporate the sparse prior: $
> \lambda\sum_{n=1}^{N}|| f_{n}([I_n^t])||_{\ell_1}$, where $\lambda$ is a sparse regularization parameter. The results with different $\lambda$ values are shown in the Table below. The results show that the sparse regularization with a suitable $\lambda$ has the potential to improve the performance of our method, while higher $\lambda$ may over-constrain the model. It would be interesting to develop other effective sparse regularizations under the OFTD framework, and we leave it to future research.
>
> Table: NRE results with different intensities of the sparse regularization.
> |$\lambda$|$0$|$0.01$|$0.03$|$0.05$|$0.07$|$0.09$|
> |-|-|-|-|-|-|-|
> |Foreman|0.084|0.081|0.079|0.078|0.078|0.080|
> |Condition|0.093|0.091|0.088|0.088|0.090|0.092|
>
> >**Q3: Minor comments: There is some formatting issue in terms of using the citations to the references. The author names are repeated twice which is better to avoid.**
>
> Thanks! We will revise the citations to avoid repetition.
>
> >**Q4: According to Theorem 1, not only the distance from new data stream, but also the neural network architecture and the choice of also plays role in forgetting. There is no discussion regarding that to give some insights in the practical settings.**
>
> Yes, the regret bound in Theorem 1 is determined by both the distance term $(I_n^t+1-i_n)$ and the constant $C_2$ that relates to the rank $r$ and neural network architectures. Nonetheless, we may treat these two terms differently. In the distance term $(I_n^t+1-i_n)$, the change of position $i_n$ reflects the change of regret bound across different positions, indicating that more distant information holds larger regret bound. The constant $C_2$ is independent of the position $i_n$ and hence serves as a fixed "coefficient" role that does not reflect the heterogeneity in regret bound across different positions. Hence we mainly focus on analyzing the position-dependent forgetting behavior induced by the distance term $(I_n^t+1-i_n)$. Still, from the construction of $C_2$, we see that smaller rank $r$ leads to smaller regret bound (i.e., less forgetting), while larger rank $r$ leads to better representation and approximation ability of the model. Hence, Theorem 1 also reveals a theoretical trade-off between the forgetting behavior and the approximation ability of the model. The ablation study for the rank $r$ is given in Table 9 of the Appendix. In our experiments, we have set $r=100$ across all cases, which was shown to strike a good balance between representation ability and forgetting.
>
> >**Q5: The parameters of the beta distribution is currently fixed, but according to the theorem, it is dependent on the learning process and can be influenced by that.**
>
> Yes, currently the Beta parameter $\beta$ is fixed. It is a great idea to dynamically adjust the Beta parameter based on the learning process. Inspired by your comments, we conduct experiments using a linear dynamic scheme for $\beta$ (i.e., $\beta_t=\beta_0+t\Delta\beta$, where $\beta_0=1$), as shown in the Table below. Better NRE results are observed when $\Delta\beta$ is a positive value (i.e., long-tail distribution). Conversely, when $\Delta \beta$ is a negative number, the gains are negative. We attribute this result to the nature of our forgetting bound, which depends on the relative position $(I_n^t+1-i_n)$ (as shown in Eq. (8) of the main text). It shows that distant information is more likely to be forgotten, and hence a long-tail memory buffer with $\Delta\beta>0$ is favored. The design of more complex $\{\beta_t\}$ schedules and their theoretical analysis are not trivial due to the complexity of the optimization problem. We leave this promising direction to future research.
>
> Table: NRE results with dynamic scheme $\beta_t=1+t\Delta \beta$ on the Foreman data.
> |$\Delta \beta$|-0.01|-0.005|0|0.005|0.01|0.015|0.02|Fix $\beta_t=1.2$|
> |-|-|-|-|-|-|-|-|-|
> |NRE|0.092|0.087|0.086|0.083|0.083|0.084|0.084|0.084|
>
> >**Q6: There are other influencing factors such as rank, the choice of omega, the number of layers. How are these factors affecting the performance and be selected for optimal outputs.**
>
> Thanks! We have tested the influence of the rank $r$ and $\\omega_0$ in Table 9 and Table 10 in the Appendix after the main text. We paste them here for your convenience. We have also tested the influence of number of layers, as shown in the Table below. OFTD is relatively robust to these parameters across suitable ranges of values. In our experiments, we set $r=100$, and select $\omega_0$ from the small candidate set $\\{0.3,1.5\\}$ to obtain satisfactory performance. The number of layers is consistently set as $3$. While the ablation studies show that further fine-tuning these parameters (such as $\omega_0$) is expected to enhance the results, our current hyperparameter settings already suffice to achieve satisfactory performances across datasets.
>
> Table: Ablation study for the rank $r$. (Average results on Beijing and Madrid data)
> |$r$ | 20|40|60|80|100|120|140|160|180|
> |-|-|-|-|-|-|-|-|-|-|
> |NRE|0.1253|0.1255|0.1243|0.1242|0.1245|0.1251|0.1235|0.1246|0.1252|
>
> Table: Ablation study for the hyperparameter $\omega_0$ in the sine activation function. (Average results on Beijing and Madrid data)
> |$\omega_0$ |0.1|0.3|0.5|0.7|0.9|1.1|1.3|1.5|1.7|
> |-|-|-|-|-|-|-|-|-|-|
> |NRE|0.1410|0.1319|0.1184|0.1168|0.1196|0.1272|0.1358|0.1430|0.1478|
>
> Table: Ablation study for the depth (number of layers) of INRs on the Condition data. We vary each factor while fixing the other ones to assess its individual effects.
> |Depth|2|3|4|5|6|7|8|9|10|
> |-|-|-|-|-|-|-|-|-|-|
> |NRE|0.099|0.093|0.088|0.086|0.085|0.088|0.089|0.096|0.109|
>
> >**Q7: I could not see much information presented regarding the baselines and their tensor decomposition methods. Are all of them CP? Or have you tried with other tensor decomposition methods such as Tucker etc?**
>
> The baseline methods ATD and SIITA are Tucker decomposition-based methods; Grouse, Grasta, and Petrels are low-rank matrix decomposition-based methods; TeCPSGD, OLSTEC, SOFIA, ACP, STF, OnlineSGD, and GOCPT are CP decomposition-based methods. Our OFTD demonstrates superior performance compared to these baseline methods using various decomposition paradigms. We will clarify these baseline settings in the main text.
>
> Should you need further information, please let us know. We look forward to hearing from your feedback!

---

> > ### Comment · Reviewer_ZeJq · 2025-08-03
> >
> > I thank the authors for the detailed response. Also, thanks for addressing most of my comments. I am raising the score accordingly.

---

> > > ### Author Response · Authors · 2025-08-03
> > >
> > > We sincerely thank the reviewer’s constructive comments and recognition of our manuscript and responses!

---

### Decision · Program_Chairs · 2025-09-17

**Decision:**

Accept (spotlight)

**Comment:**

This paper proposes OFTD, a novel online functional tensor decomposition framework that integrates implicit neural representations with continual learning and memory replay to address streaming tensor completion. Reviewers found the work well-motivated, technically sound, and clearly written, highlighting its originality in bridging tensor decomposition with INR-based continual learning. The rebuttal successfully resolved all raised concerns, leading to positive consensus among reviewers. The work’s solid theory, strong experiments, and potential impact make it stand out, and thus it is recommended for Spotlight.